# Charting the Frontier: How Optimizing Performance Yields Accurate Scaling Laws on a Shoestring

## Abstract

Predicting model performance at larger scales enables the design of training strategies and architectures tailored to specific performance targets. Empirical scaling law research identifies functional forms to aid this prediction task. These describe the relationship between loss and compute using a loss-compute frontier defined by learning curves. Due to the empirical nature of this approach, the computational burden is substantial, making strategic resource allocation essential – yet it remains surprisingly underexplored. In this work, we address this shortcoming by exploring the suitability of Successive Halving (SH) and SH combined with parametric and non-parametric surrogate models. In addition to enabling a more systematic allocation of a given compute budget, our findings show that SH paired with surrogate models yields a set of learning curves that includes one with a lower loss-compute value than what naive uniform allocation or an SH-only approach can obtain. Our experiments demonstrate mean relative improvements of up to $2.84\%$ and $5.47\%$ on real-world and synthetic learning curve datasets. This strategic resource allocation enables us to obtain accurate scaling laws at significantly reduced computational costs, saving up to $98.7\%$ over the traditional exhaustive approach.

## 1 Introduction

Recent trends in Machine Learning have moved towards larger and increasingly data-hungry models. This is particularly the case for Large Language Models (LLMs), which can be prohibitively expensive to train and test, hindering their ability to be explored and tuned in a systematic way. To address this problem, *performance prediction* has become an important technique to facilitate decision making for the deployment of large-scale ML systems (Xia et al., 2020; Ye et al., 2021; Schram et al., 2023). This includes not only model-based decisions (such as hyperparameter settings) but also exploration of data requirements and out-of-domain performance evaluation.

Learning curves are an important source of information. In the context of LLMs, they have been used to develop *scaling laws*: functional forms which are fitted to empirically obtained learning curves, which describe the performance of an LLM given a specific set of scales, such as computational resources, model capacity and dataset size (Kaplan et al., 2020; Hoffmann et al., 2022). Estimating a scaling law for a new family of models can be a valuable resource for decision making and deployment – but is extremely cost intensive. Recent studies have required training hundreds of models across different parameter sizes and training durations (Kaplan et al., 2020; Hoffmann et al., 2022). However, training all models to full convergence is computationally wasteful and expensive, making such studies infeasible in budget-constrained settings. Despite this, scaling laws remain a valuable tool for targeted budget investment and decision making, highlighting the need for more efficient and systematic approaches – a topic that remains surprisingly underexplored.

In the area of hyperparameter optimization, however, optimal resource allocation is well studied. As finding an appropriate model architecture by optimizing over the hyperparameters is a costly process in terms of time and computational resources, automatic hyperparameter optimization methods aim to alleviate the contributing factors, including large numbers of hyperparameters, non-linear hyperparameter interactions, and non-convex non-differentiable optimization problems. A variety of parametric and non-parametric methods have been successfully employed via Bayesian optimization

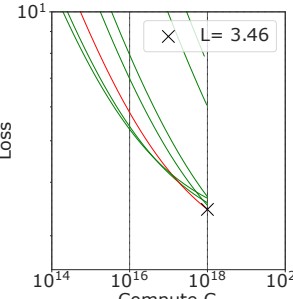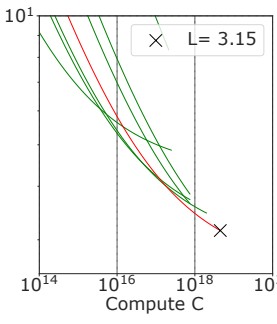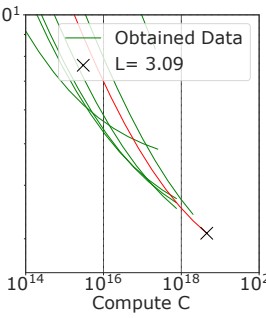

Figure 1: Comparison of different compute budget allocation strategies using the synthetic LC dataset. Approaches are described in Section 3. **Left:** Conventional uniform allocation. **Middle:** Successive Halving. **Right:** Successive Halving with multitask Gaussian Process (LMC) as surrogate model.

to obtain the best hyperparameter settings more efficiently by predicting learning curve trends (Swersky et al., 2014; Klein et al., 2016; Kadra et al., 2023).

In this paper, we build upon some of these insights and explore cost-effective approaches to obtain compute-loss frontiers for scaling law research in budget-constrained experimental settings. Specifically, we set out to provide insights to the following two questions: 1) Can strategic budget allocation help to obtain accurate scaling laws while significantly reducing the required computational budget? 2) If so, can we further improve such allocation by leveraging characteristics of the particular curve shapes that are inherent to model training? To provide answers to these questions, we build our experiments around a budget-based hyperparameter optimization method called Successive Halving (SH) (Jamieson & Talwalkar, 2016). We then further extend this approach by combining it with two probabilistic methods: a multitask Gaussian Process model, and Deep Ensemble models – both tailored to predict learning curves. Given a family of models with no prior information and a fixed computational budget, we use our methods to strategically allocate resources to a set of models of various sizes to obtain their learning curves. We find that compared to a conventional uniform budget allocation and our SH-only approach, using Successive Halving in combination with a surrogate model allows us to obtain a set of learning curves that includes one with a lower loss-compute value than can otherwise be obtained (see Figure 1). In addition, the probabilistic nature of our chosen surrogate models not only allows us to obtain accurate scaling laws at significantly reduced cost, but can further provide upper and lower limits on the scaling laws via the models' confidence bounds.

## 2 RELATED WORKS

**Learning Curves.** While validation performance is the main metric that guides model tuning and performance prediction, learning curves (LCs) can provide valuable additional information for these tasks. In Bayesian optimization, Swersky et al. (2014) employed a surrogate model using a Gaussian Process (GP) with an exponential decay kernel which proved beneficial to predict the behavior of LCs. Klein et al. (2020) used a similar approach for neural architecture search, while Lin et al. (2024) used product kernels for scenarios with missing values. Besides GPs, other surrogate models have been explored as well, such as parametric functions (Domhan et al., 2015), Bayesian neural networks (Klein et al., 2017) and ensembles of networks (Kadra et al., 2023). Our work uses surrogate models to predict LC trends of models of various sizes.

**Hyperparameter Optimization Techniques.** Much of the work in hyperparameter optimization relies on sequential methods like Bayesian optimization. Among those, the one most related to our work is by Swersky et al. (2014), where prior information in the form of an exponential decay kernel is leveraged. Another stream of works focuses on parallelizable methods. Among those, decision-theoretic methods define a hyperparameter search space and determine the best-performing hyperparameter combination by performing either an exhaustive search using grid-search methods or a random selection using random search methods (Bergstra et al., 2011; Bergstra & Bengio, 2012). Taking this one step further, multifidelity optimization algorithms such as Hyperband additionally allocate limited resources in each iteration and eliminate poorly performing hyperparameter combinations to save computation time (Li et al., 2018). Each iteration considers a different resource allocation strategy; and in combination with Bayesian optimization, more informed decisions based

on previously-explored configurations can be made (Wang et al., 2018). Hyperband calls Successive Halving (SH) (Jamieson & Talwalkar, 2016) as a subroutine. SH is a non-stochastic, bandit-based optimization method that finds the best configuration which optimizes a metric of choice. It applies early stopping in each round, then reduces the number of candidate configurations by a predefined factor. Concurrently, it efficiently allocates a considered resource in each iteration, e.g., the number of training iterations; starting from a low-resource many-configurations setup, and ending with the most promising configuration and the largest amount of resources allocated. Our work combines SH with surrogate models to allow an informed resource allocation strategy early on. While our work is related to (Swersky et al., 2014) and (Kadra et al., 2023), we instead focus on parallelizable methods. Furthermore, our scenario requires appropriate resource allocation to all models of interest to obtain a dense compute-loss frontier. And while we build upon insights and strategies from hyperparameter optimization techniques, our problem is one of resource allocation for a dense loss-compute frontier.

**Scaling Laws.** Given the increasing interest in the capabilities of LLMs, scaling laws (SLs) have arisen as a way to extrapolate the behavior of these models given a set of LCs (Kaplan et al., 2020). These laws can offer several benefits, including improved interpretability of neural networks (Bahri et al., 2024; Abnar et al., 2022; Bansal et al., 2022), more effective training sample size planning and a reduced carbon footprint (Domhan et al., 2015; Johnson et al., 2018). While theoretical approaches have been developed to facilitate the prediction of SLs for only a limited number of architectures (Hutter, 2021; Sharma & Kaplan, 2022), most successful methods for predicting scaling behavior across a wide range of networks are based on empirical studies (Henighan et al., 2020; Hägele et al., 2024; Hoffmann et al., 2022; Ghorbani et al., 2022; Hestness et al., 2017; Rosenfeld, 2021). This, in turn, motivates cost-effective approaches to obtain LCs for accurate SL estimation. Please see Appendices A and S for extended discussions.

**Efficiency in Scaling Law Research.** Hägele et al. (2024) proposed an approach to reduce the computation requirements to obtain SLs by using alternative model training techniques: constant learning rates with cooldowns and stochastic weight averaging. Our work is orthogonal to this approach, and is focused on optimizing the computational budget allocation for training. Several other recent works have focused on efficiently deriving scaling insights across and/or within different model families, mainly based on results reported for various benchmarks. Ruan et al. (2024) pioneer this direction of 'observational' or 'benchmark scaling laws', utilizing dimensionality reduction via PCA to demonstrate that model performance can be modeled in a low-dimensional space of abstract 'principal capabilities'. Their method is primarily designed to interpret the skills of already trained LLMs and predict complex downstream performance based on simpler proxy metrics. Building on this latent-variable perspective, Polo et al. (2025) propose a conceptually similar but more parametric framework in which benchmark performance is modeled by a small set of explicitly interpretable 'skills' (e.g., reasoning). The authors also specifically design their approach to predict the performance of 'future', yet untrained models across benchmarks and model families. A somewhat orthogonal line of work by Choshen et al. (2025) provides a comprehensive guide to improving the robustness of traditional loss scaling law estimation by analyzing and mitigating the effects of variability across training regimes and model families within existing learning curves. However, these methods focus on extracting insights about scaling behavior from existing evaluation or training data; whereas our work in contrast focuses on optimizing the allocation of compute to actively generate the underlying learning curves themselves.

## 3 STRATEGIC BUDGET ALLOCATION FOR COST-EFFICIENT SCALING LAWS

Starting with a model set $\mathcal{M}$ drawn from a family of interest (e.g., Transformers) and seeking to derive a scaling law, we are limited by a finite *total* computational budget $B$ for training. The central challenge is to allocate this budget effectively to select models that contribute to the compute-optimal frontier and hence the scaling law, while minimizing 'wasted' effort on less promising candidates.

We define a *learning curve* $L(C)$ as the validation loss $L$ of a model given its compute $C$. Depending on its number of parameters $N$ and dataset size $D$, each model needs a different amount of compute for one training iteration. Figure 2 shows such learning curves for nanoGPT models of various sizes in green. Note that the model size is indirectly represented as the starting point and early slope of each curve: Smaller models reach lower loss values earlier, but also plateau faster – and are hence located more to the left.

In theory, $L(C)$ is defined over $[0, \infty]$. In practice, however, we obtain only a partial curve for each model $m \in \mathcal{M}$, with its range limited by the model's maximum allocated compute budget $C_m$. Using the available set of models $\mathcal{M}$, we analyze the corresponding set of learning curves to derive a scaling law (SL). Among the various approaches discussed by Hoffmann et al. (2022), we focus on the one most relevant to this work, which is to fit the following functional form to the loss-compute frontier in a compute region of interest (Pearce & Song, 2024; Kaplan et al., 2020): $L^{\mathrm{SL}}(C) = (C/\alpha)^{-\gamma}$, where $\alpha$ and $\gamma$ are parameters to be optimized.

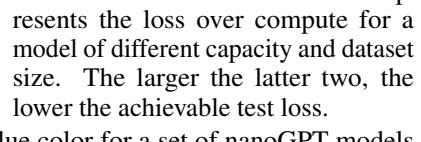

Figure 2: A set of LCs obtained from nanoGPT models. Each curve represents the loss over compute for a model of different capacity and dataset size. The larger the latter two, the lower the achievable test loss.

Scaling laws depend on both the compute range of interest and on the particular set of models being analyzed. A set of very large models will lead to a different law than a set of small models as discussed in Hoffmann et al. (2022, Figure A5) and also shown in Appendix I. We therefore assume that we are interested in a predefined fixed compute range, which aligns well with real-world budget-constrained scenarios. Figure 2 illustrates the obtained scaling law in blue color for a set of nanoGPT models using a log-log scale, meaning the scaling law has a linear form. The compute-efficient frontier is here defined for a compute range of interest between $10^{18}$ and $10^{20}$, and is visualized in black.

In addition to the previously detailed influences, the total compute budget $B$ that is available to train the set of models will also affect the scaling law. Theoretically, an infinitely large budget could be used to train each model until convergence – however, might result in arbitrarily (and unnecessarily) large amounts of compute $C_m$ assigned to each model $m$. In contrast, considering a more realistic setup of having a limited total compute budget $B$, each $C_m$ needs to be *chosen* such that the compute will be distributed among all models in $\mathcal{M}$; and should ideally be limited for those models early in the process whose learning curves will not be able to contribute to the loss-compute frontier. This leads us to the core question: *How can we achieve this in practice*, i.e., choose $C_m$ for each model to satisfy these expectations but without unnecessarily wasting precious computational resources?

We tackle this question in two distinct steps. Note that since $C_m$ represents the cost of obtaining the learning curve for a model $m \in \mathcal{M}$, we can define the total cost of obtaining all curves for the entire set of models as $C_{\mathcal{M}}$, with $C_{\mathcal{M}} = \sum_{m \in \mathcal{M}} C_m$. Now let us assume we could measure the 'goodness of fit' $G(\mathcal{M})$ of a scaling law for this set. Then, given a total budget $B$, our objective of strategically allocating computational budget to each model $m$ is defined as

$$\underset{C_1,\ldots,C_m}{\arg\max}\ G(\mathcal{M})\ \text{ s.t. }\ C_{\mathcal{M}} \le B. \tag{1}$$

There are two major challenges in solving this problem. First, defining $G(\mathcal{M})$ in a principled way would require having a gold-standard SL to compare with. However, this gold-standard could only be obtained by training all available models until convergence, which is time-consuming, computationally expensive and infeasible in budget-constrained research settings. Second, even if we had $G(\mathcal{M})$, this problem would likely be intractable without making assumptions about this metric. To overcome this and still obtain an approximate solution for Equation (1), we first solve a proxy problem instead: allocating budget across the set of models in pursuit of finding the model that reaches minimum validation loss within our budget constraint

$$\underset{m \in \mathcal{M}}{\arg\min}\ L_m(C)\ \text{ s.t. }\ C_{\mathcal{M}} \le B. \tag{2}$$

The budget constraint makes the problem non-trivial: larger models have the potential to reach smaller loss values, but might exhaust the budget before this happens. The constraint still leads to an intractable problem, but the objective function is readily available, meaning we can explore approximate methods to solve the problem. Importantly, optimizing Equation (2) will generate a set of curves *as a byproduct*, which we then use to approach Equation (1) empirically to find a good scaling law.

**Successive Halving w/ and w/o Surrogates.**
We approach this challenge by building upon the Successive Halving (SH) algorithm (Jamieson & Talwalkar, 2016) that is commonly used to find the 'best' set of hyperparameters. In contrast to this original objective, our main goal is *not* to ultimately identify the single best-performing model as a function of the compute – rather, we aim to obtain a *set* of near-optimally trained models within the available compute. We utilize two different variants of SH to address this, as introduced in the following. Starting with a set of models $\mathcal{M}_{r=0}$, each model in this set is allocated a fixed 'starting budget' $C_0$ and is trained until this budget $C_0$ is exhausted (round $r = 0$ in Algorithm 1). At this point, our two approaches

---

**Algorithm 1** Strategic Compute Allocation

**Input:** $B$: Total compute budget in FLOPs.
$\mathcal{M}_{r=0}$: Set of models in round $r = 0$.
$\eta$: Pruning parameter.
**Output:** $\mathcal{L}$: Set of learning curves.
**Initialize:** Models $m \in \mathcal{M}_0$.
1: **for** $r \leftarrow 0$ to $\lceil \log_\eta |\mathcal{M}_0| \rceil - 1$ **do**
2: $\quad C_r \leftarrow \left\lfloor \frac{B}{|\mathcal{M}_r| \cdot \lceil \log_\eta |\mathcal{M}_0| \rceil} \right\rfloor$
3: $\quad \tilde{\mathcal{L}} = \{ \texttt{obtain\_LCs}(m, C_{0:r}) : m \in \mathcal{M}_r \}$
4: $\quad \mathcal{M}_{r+1} = \texttt{Top\_k}(\mathcal{M}_r, \tilde{\mathcal{L}}, \eta)$
5: $\quad \mathcal{L} \leftarrow$ Update by $\tilde{\mathcal{L}} \setminus \hat{\mathcal{L}}$.
6: **end for**
$\quad$ **Return:** $\mathcal{L}$.

---

diverge. The original SH algorithm now simply retrieves the learning curves for all $m \in \mathcal{M}_r$ (line 3) and selects the most promising models based on their *current* validation losses (line 4). The selected models continue to the next round, where they are assigned fresh budget to continue training, while the discarded ones are kept 'as-is'.

In contrast, our extended SH approach employs a *surrogate model* that uses the observed learning curves up to this point to predict their likely continuation. Specifically, the surrogate estimates the lowest achievable *future* loss for each model in $\mathcal{M}_r$, assuming it successfully advances through all remaining rounds. Based on these *predicted future* losses, the top-$k$ most promising models are selected to proceed to the next round (line 4) and receive additional training budget, while the rest are kept 'as-is'. The detailed algorithmic description of all steps is provided in Appendix B.

Note that while this process solves the proxy task (Equation (2)) – namely, selecting the most promising models in each round – the iterative and strategic allocation of budget results in the full model set $\mathcal{M}$, where each model was trained as long as it was considered likely to contribute to the loss-compute frontier. In this way, our method approximates the ideal allocation objective described in Equation (1), and allows us to obtain a scaling law from the resulting set of learning curves $\mathcal{L}$.

**Linear Model of Co-regionalisation (LMC).** We investigate the suitability of two different types of surrogate models in this work. Our first surrogate formulates the learning curve extrapolation task as a multi-input, multi-output problem. Given that learning curves appear to exhibit correlated behavior (e.g., Figure 3), we employ Latent Gaussian Processes to capture the correlations among outputs, with the objective of improving predictive accuracy. The likelihood of all observations $\mathbf{Y} \in \mathbb{R}^{Q \times N}$, where $Q$ denotes the number of outputs and $N$ the number of observations per output, is modeled as $\mathbf{Y} \sim \mathcal{N}(\mathbf{0}, \mathbf{\Sigma} + \sigma^2 \mathbf{I})$, assuming independent and identically distributed (i.i.d.) Gaussian observation noise with variance $\sigma^2$. Our proposed LMC has the following kernel

$$\mathbf{\Sigma} = \mathbf{B}_1 \otimes \mathbf{K}_{\text{ExpDec}} + \mathbf{B}_2 \otimes \mathbf{K}_{\text{White}} + \mathbf{B}_3 \otimes \mathbf{K}_{\text{Bias}}, \quad \text{with } \mathbf{B}_j = \mathbf{W}_j^T \mathbf{W}_j + \text{diag}(\kappa_j),$$

and is composed of three different individual sub-kernels, each modulated via a matrix $\mathbf{W}_j$ which maps the latent function values to the observed data outputs. The first sub-kernel $\mathbf{K}_{\text{ExpDec}}$ denotes an exponentially decaying kernel that models the characteristic shape of the learning curves (Swersky et al., 2014), and we constrain $\mathbf{W}_1$ and $\kappa_1$ to be positive to ensure an appropriate curvature. $\mathbf{K}_{\text{White}}$ is a white kernel accounting for any i.i.d. additive white Gaussian noise present in the data. To model the individual mean component of each learning curve, we incorporate $\mathbf{K}_{\text{Bias}}$. We fix its associated parameters to $\mathbf{W}_3 = 0$ and $\text{var}_{\text{Bias}} = 1$, such that the mean is governed by a positively constrained parameter $\kappa_3$. This ensures that the resulting mean values across all learning curves remain strictly positive. More detailed information for this surrogate model is provided in Appendix F.1.

**Deep Ensemble (DE) Methods.** As an alternative to our non-parametric LMC surrogate model, we also explore a parametric approach in the form of Deep Ensemble methods for loss-compute curve extrapolation (Kadra et al., 2023). A single shared function across all model sizes $N$ is fitted by conditioning its coefficients on the model size $N$ (Appendix F.2 for details). We consider three functions that are known to be suitable to regress and extrapolate learning curve shapes (Viering & Loog, 2022); namely the power law (PL) function, the exponential function (EXP) and the the Morgan-Mercer-Flodin (MMF) function, which are defined in detail and illustrated in Appendix F.2.

## 4 EXPERIMENTS

We conduct a number of experiments to demonstrate the effectiveness of our approach and provide key insights into its performance and potential practical implications. All code used in the experiments will be made publicly available. Our GP-based surrogates are optimized using L-BGFS with 20 random restarts to avoid local optima, and are implemented using GPflow (Matthews et al., 2017). We use ensembles of 5 randomly-initialized two-layer perceptrons following (Kadra et al., 2023), and set the number of iterations to 1000. All experiments are averaged over 100 runs unless otherwise stated. We report the mean relative improvement compared to the non-surrogate SH's minimum loss as $(l^*_{\text{SH},r} - l^*_{\text{Surrogate},r})/l^*_{\text{SH},r}$ for all runs where SH cannot obtain the optimal solution, and the maximum relative improvement is given in parentheses. Additional results and metrics can be found in the appendices. We start our analyses with the non-parametric surrogate method SH LMC.

### 4.1 STRATEGIC BUDGET ALLOCATION AND THE BENEFIT OF SURROGATE MODELS

In the first set of experiments, we evaluate the effectiveness of our proposed strategic budget allocation via both Successive Halving (SH) and SH with the non-parametric LMC surrogate model in solving the proxy task of obtaining a set of well-trained models that yield low validation loss.

**Setting expectations.** Due to the nature of our problem, we expect the SH algorithm in its original form without integrated surrogate prediction to inherently favor smaller models in each round $r$, especially early on: Small models generally need less compute to fulfill one training iteration and reach smaller loss values earlier (see Figures 1 and 3). However, they usually also plateau earlier and are therefore naturally not able to achieve a loss value as small as a bigger model could potentially achieve later on – rendering this naive form of the SH algorithm suboptimal for our scenario.

**First qualitative insights.** A side-by-side illustration of the budget allocation approach using SH and SH LMC is shown in Figure 1. As the same compute is allocated to all models in each round $r$, the corresponding learning curves stop at the same compute value. As detailed earlier, only the 'most-promising' models are selected for further training in each round, leading to learning-curves with different end-compute values (here between $10^{16}$-$10^{18}$) depending on the number of rounds the models participated in. Note that predicting the future continuation of a learning curve via the surrogate while taking both the model $m$ and its maximally remaining budget into account leads to a *different model selection* (highlighted in red in Figure 1), ultimately achieving a lower loss value. In essence, combining SH with the surrogate model to predict a model's *future* performance is clearly beneficial to make informed and strategic decisions about budget allocation – particularly when their *current* performance might not yet reflect their true potential. We now go on to quantitatively validate these insights across different synthetic and real-world learning-curve (LC) datasets.

**Evaluating on synthetic data.** Loss-compute LCs can be created synthetically by leveraging the formula for the $L(N, D)$ scaling law (SL) in combination with the approximation $C = 6ND$ (Hoffmann et al., 2022; Kaplan et al., 2020) for given $N$ and $C$ values. $N$ represents the number of model parameters of each model, i.e., the model size, and $C$ the available compute budget. More information is given in Appendix H. When creating this dataset, we use the coefficients according to Hoffmann et al. (2022).

The experiments are conducted in log-compute and log-loss space. The number of parameters $N$ for each model in initial set of models $\mathcal{M}_0$ is randomly sampled from the parameter

Table 1: A comparison of the performance of Successive Halving (SH) LMC and Uniform Allocation (UA) to the original Successive Halving (SH) for the synthetic dataset. The average is taken over 100 runs. In each run models were selected randomly. We assume a fixed computational budget $B$ in petaFLOPs for a number of models $M_0 = |\mathcal{M}_0|$.

| | | SH | SH LMC | UA |
|---|---|---|---|---|
| $M_0$ | $B$ | mean loss | mean (max), rel. improv. | mean (max), rel. degradation |
| 5 | $10^2$ | $6.40 \pm 9.07$ | $5.15\%$ (20.30%) | $-10.17\%$ (−24.70%) |
| 5 | $10^3$ | $4.62 \pm 3.10$ | $4.63\%$ (7.57%) | $-9.00\%$ (−25.52%) |
| 5 | $10^4$ | $3.84 \pm 2.03$ | $5.47\%$ (16.70%) | $-7.59\%$ (−22.24%) |
| 10 | $10^2$ | $4.73 \pm 0.49$ | $4.01\%$ (17.78%) | $-15.54\%$ (−28.70%) |
| 10 | $10^3$ | $3.86 \pm 0.38$ | $2.38\%$ (6.11%) | $-14.06\%$ (−33.01%) |
| 10 | $10^4$ | $3.26 \pm 0.29$ | $4.02\%$ (13.63%) | $-11.71\%$ (−32.69%) |
| 20 | $10^2$ | $4.69 \pm 0.10$ | $1.69\%$ (9.40%) | $-22.73\%$ (−45.41%) |
| 20 | $10^3$ | $3.80 \pm 0.09$ | $1.56\%$ (4.89%) | $-19.47\%$ (−33.07%) |
| 20 | $10^4$ | $3.18 \pm 0.09$ | $1.50\%$ (6.53%) | $-16.40\%$ (−32.27%) |

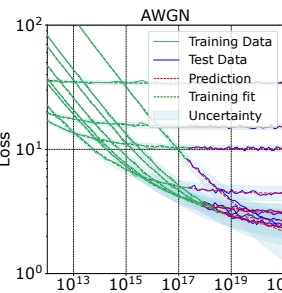
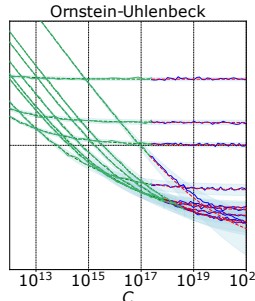
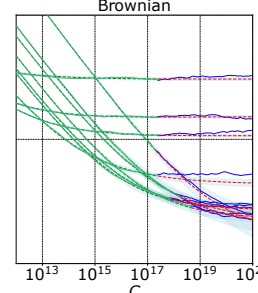

Figure 3: Illustration of LMC extrapolation during SH LMC for various noise models. Green solid lines: Performance of a trained model (used for LMC training). Red dashed lines: GP extrapolation.

space $[2^2, 2^3..., 2^{42}]$ to cover a wide range of model sizes. All SH LMC models are trained using 20 datapoints extracted from each LC.

Table 1 shows the mean (max) relative improvement of Successive Halving (SH) with integrated LMC surrogate model compared to conventional SH for the learning curves of 5, 10 and 20 models across budgets of $10^2$, $10^3$ and $10^4$ petaFLOPs. We also report the mean (max) 'relative degradation' in performance that is faced when using a traditional non-strategic uniform budget allocation (UA) instead of SH. The results indicate that both Successive Halving and SH with LMC clearly outperform the naive uniform allocation strategy, with with surrogate-based SH LMC performing better than plain SH as reflected by the positive mean (max) relative improvement. This is especially pronounced for setups with comparably few models (5 and 10), where SH LMC can leverage its advantage: The multitask Gaussian Process learns the extrapolation trends required to correctly judge a model's future benefit from related 'tasks' (i.e., all learning curves in our set). In this way, small and fast-converging models can provide information when 'an inclination' is to be expected – knowledge that can then be leveraged to predict the learning curve trend of larger models, which might at the current timestep *not yet* seem particularly promising (and are hence rejected by conventional SH); just as we expected.

**Performance under noisy conditions.** Given that real-world data is rarely as clean as synthetic data, we test the robustness of SH LMC by adding weighted noise to the data, assuming the following three noise models: White Gaussian noise, Brownian noise and Ornstein-Uhlenbeck noise. All sampled noise values are added to the logarithmic loss. Note that while Brownian noise will not resemble the learning curve behavior in the long term (given its inherent properties), it is still a valuable noise model to challenge the extrapolation capabilities of our surrogate model in the short term.

Closer inspection of the LC extrapolation results visualized in Figure 3 shows that SH LMC decided to allocate more budget to large models across all three noise models, based on the surrogate's prediction of these models' ability to *eventually* achieve lower loss values (within the maximum available budget) – a strategic decision based on information that is not available to SH without surrogate. The direct comparison of the minimum attained loss shown in Figure 4 further shows that this seems to hold across all three noise models and varying noise intensities: Employing 5 or 20 learn-

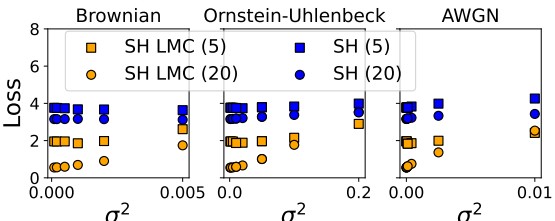

Figure 4: Average minimum loss obtained by SH and SH LMC in noisy scenarios. Reporting results for 5 and 20 learning curves (square and circle). Noise level denoted by $\sigma^2$, which represents the noise intensity that was steadily increased to generate the noise values.

ing curves, SH LMC (orange) consistently outperforms SH (blue) by finding a smaller minimum average loss value (averaged over 30 runs). More examples are provided in Appendix J.

**Real-world nanoGPT experiments.** To further demonstrate the efficacy and benefit of our strategic allocation approaches on real-world data, we train a variety of nanoGPT models (Karpathy, 2022) of up to $1.5B$ parameters – the same maximum number of parameters used in seminal scaling-law paper Kaplan et al. (2020) – and use their learning curves as basis to contrast Successive Halving (SH) and SH LMC to three different parametric Deep Ensemble (DE) surrogate alternatives. Our created dataset comprises 90 learning curves corresponding to models with the number of layers ranging

from 8 to 48, and embedding dimensions varying between 512 and 1600 – with the model parameter space $N$ defined by the combination of both architectural dimensions. All models are trained on the 10B token split of the fineweb-edu dataset (Penedo et al., 2024). More details regarding the full set of learning curves and minor preprocessing steps are visualized and detailed in Appendix M. We refer to this set of learning curves as the 'nanoGPT dataset' in the following. All surrogate models are trained using 20 datapoints extracted per curve, and total budgets of $10^4$ and $10^5$ petaFLOPs are allocated – which are chosen slightly higher than the ones used for the synthetic results to reflect the increased model size and hence computational resources used in these experiments.

As shown in Table 2, all strategic allocation methods continue to outperform naive uniform allocation (UA). Successive Halving with the LMC surrogate is able to improve upon SH by up to $2.84\%$. SH DE methods also perform well, but seem slightly less capable to model the noisy curve shapes in the real-world dataset. Compared to the synthetic data, the improvements obtained by SH LMC seem more consistent across varying numbers of models of interest $M_0$, but overall slightly smaller. One possible explanation is the difference in dataset structure: learning curves in the nanoGPT dataset are located in much closer proximity to each other (Figure M.1 in Appendix M), closely resembling the data shown in Kaplan et al. (2020, Figure 1, left) – which increases the need for highly accurate predictions of curve extrapolations. Nevertheless, note that given the increased cost of training large models, even minor improvements can still yield significant cost savings for obtaining a representative set of learning curves that is well-suited to determine scaling laws.

## 4.2 FROM PROXY TASK TO SCALING LAW

Having demonstrated the suitability of both Successive Halving (SH) and SH with surrogate models to solve our proxy task and strategically obtain a set of learning curves, we are now facing the question how well this translates to obtaining the scaling law for the involved models. However, evaluating this is slightly more complicated than appears at first: Scaling Laws (SLs) are traditionally obtained empirically and depend on many factors, and there exists no 'one gold-standard ground truth' value (Appendix I). To nevertheless provide a basis for evaluation, we derive two different 'ground truth' SLs for the nanoGPT dataset as follows: The first one is the scaling law obtained using the *whole dataset* (i.e., all 90 learning curves), referred to as *Full Data SL*. The second 'ground truth' is the law obtained when all selected models $m \in \mathcal{M}_0$ for a particular experiment are fully trained until convergence, referred to as *Entire LCs SL*. Please refer to Appendix R for more details.

We define the compute region of interest for these experiments as ranging from $10^{18}$ to $10^{20}$ FLOPs. To quantify the deviation between an obtained scaling law (SL) and the ground truth SLs, we compute the Area between the Curves (AbC), with $AbC \in (0, \infty)$, over this region. A visualization of the AbC metric is provided in Appendix Q. Smaller AbC values indicate a closer match between the obtained and ground truth scaling laws. We compare the scaling laws for the learning curves obtained by Successive Halving (SH) and SH LMC, which we chose as representative surrogate method given it previously outperformed all SH DE variants on the nanoGPT dataset. Table 3 shows that *both* strategic budget allocation approaches yield scaling laws that closely approximate the ground truths, demonstrating their effectiveness in this setting. SH LMC achieves slightly lower average loss and hence regret, consistent with earlier experimental results, but the overall similarity between the two approaches highlights that both are well suited for the task. More importantly, both methods operate

Table 2: Performance comparison of Successive Halving (SH) with SH LMC and Deep Ensembles approximating the power law (PL), exponential (EXP) and MMF functions. Models were selected at random in each of the 100 runs. The maximum improvement in percent is shown in parentheses.

| $M_0$ | $B$ | SH mean loss | SH LMC mean (max) rel. improv. | SH DE PL mean (max) rel. improv. | SH DE EXP mean (max) rel. improv. | SH DE MMF mean (max) rel. improv. | UA mean (max) rel. degradation |
|---|---|---|---|---|---|---|---|
| 5 | $10^4$ | $3.17 \pm 0.06$ | $\mathbf{2.58}\%$ (6.90%) | $1.64\%$ (4.36%) | $2.32\%$ (5.45%) | $1.79\%$ (4.36%) | $-5.09\%$ (−10.33%) |
| 5 | $10^5$ | $2.97 \pm 0.03$ | $2.36\%$ (6.92%) | $\mathbf{2.40}\%$ (5.66%) | $2.15\%$ (5.38%) | $2.11\%$ (5.38%) | $-0.74\%$ (−2.73%) |
| 10 | $10^4$ | $3.23 \pm 0.05$ | $\mathbf{2.42}\%$ (5.22%) | $2.16\%$ (10.01%) | $1.54\%$ (3.44%) | $1.64\%$ (3.65%) | $-7.92\%$ (−21.59%) |
| 10 | $10^5$ | $3.00 \pm 0.02$ | $\mathbf{2.82}\%$ (6.26%) | $1.56\%$ (3.93%) | $1.91\%$ (5.10%) | $2.14\%$ (4.46%) | $-0.81\%$ (−2.66%) |
| 20 | $10^4$ | $3.30 \pm 0.02$ | $\mathbf{2.84}\%$ (4.30%) | $2.02\%$ (4.58%) | $1.35\%$ (3.26%) | $1.80\%$ (3.26%) | $-11.46\%$ (−19.59%) |
| 20 | $10^5$ | $3.03 \pm 0.01$ | $\mathbf{2.24}\%$ (6.54%) | $1.10\%$ (3.13%) | $1.44\%$ (3.13%) | $1.37\%$ (5.17%) | $-2.96\%$ (−3.75%) |

under a total budget $B$ that is *significantly smaller* than the cost of training all models individually (see second-last column), yet still produce accurate scaling laws. This demonstrates that strategic budget allocation, whether via SH or SH LMC, enables efficient approximation of scaling behavior at a fraction of the computational cost – saving up to $98.70\%$ compared to exhaustive model training.

### 4.3 EXTENDING BEYOND THE COMPUTE RANGE USING SURROGATE MODELS

Having established that both strategic budget allocation methods, Successive Halving (SH) and SH with surrogate predictions, are well-suited to efficiently obtain scaling laws, we now ask whether the unique predictive capabilities of the surrogate model can be leveraged to gain additional advantages beyond what is possible with standard SH. To explore this direction, we construct a second synthetic dataset of learning curves (LCs) for the models in Kaplan et al. (2020), following the setup from Pearce & Song (2024, Figure 7). The parameters of the underlying equations are taken from Besiroglu et al. (2024) (see also Appendix H). Although these LCs are simulated, they are derived from empirical patterns observed across many real experiments (20 Transformer architectures, varying parameter counts) and thus serve as a realistic proxy for real-world data, while providing a reliable 'ground truth' scaling law (SL). The compute region of interest is defined between $10^{14}$ and $10^{20.7}$ FLOPs, consistent with Pearce & Song (2024), and covers the compute-efficient frontier of the SL.

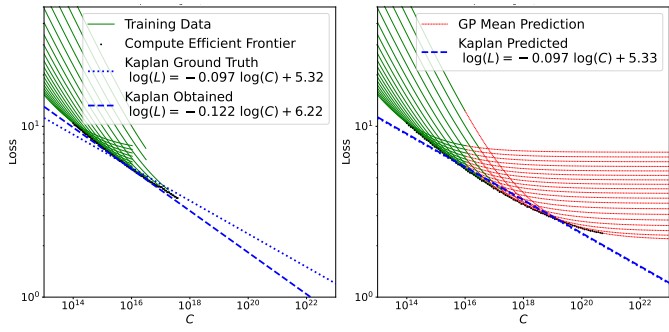

Figure 5: **Left**: SL after SH LMC. **Right**: SL after LMC extrapolation.

Figure 5 illustrates the effect of surrogate-based extrapolation. The SL obtained directly from the LCs after compute allocation via SH LMC (left) still deviates from the ground truth, whereas incorporating an *additional* GP prediction step *after* resource allocation (right) significantly reduces this gap. Table 4 compares the SLs obtained using (i) standard SH LMC (AbC SH LMC), and (ii) SH LMC followed by additional *extrapolation* via the GP surrogate (AbC GP Mean). As the available budget increases, the GP-based extrapolated SLs become more accurate, resulting in consistently lower AbC values compared to relying on post-allocation LCs alone. This predictive signal enables a more confident and accurate identification of the underlying SL, as also seen in Figure 5 (right).

In addition to the GP mean prediction, we also evaluate SLs using the Upper and Lower Confidence Bounds (UCB/LCB) provided by the GP. These bounds reflect the uncertainty in the GP's predictions:

Table 3: Area between Curves (AbC) between the defined ground truth SLs and the obtained ones after budget allocation according to SH and SH LMC. The regret on loss shows how close the respective method (SH or SH LMC) was on average to the minimum loss that could be obtained if all models $m \in \mathcal{M}_0$ were fully trained. Additionally, we show how much computing the entire LCs costs. The experiments are averaged over 100 runs, using the real-world nanoGPT dataset.

| Method | $M_0$ | $B$ | ↓ AbC Full Data SL | ↓ AbC Entire LCs SL | ↓ Loss Regret | Average Cost Entire LCs | Saving in Cost $B$ vs. Entire LCs |
|---|---|---|---|---|---|---|---|
| SH | 5 | $10^4$ | **0.09 ± 0.05** | **0.10 ± 0.06** | 0.43 ± 0.09 | $1.9 \cdot 10^5$ | 94.00% |
| SH LMC | 5 | $10^4$ | 0.11 ± 0.07 | 0.12 ± 0.08 | **0.41 ± 0.10** | $1.9 \cdot 10^5$ | 94.00% |
| SH | 5 | $10^5$ | 0.19 ± 0.10 | 0.14 ± 0.07 | 0.33 ± 0.03 | $4.1 \cdot 10^5$ | 75.61% |
| SH LMC | 5 | $10^5$ | **0.18 ± 0.09** | **0.12 ± 0.07** | **0.32 ± 0.04** | $4.1 \cdot 10^5$ | 75.61% |
| SH | 10 | $10^4$ | **0.07 ± 0.02** | **0.05 ± 0.04** | 0.56 ± 0.07 | $4.0 \cdot 10^5$ | 97.50% |
| SH LMC | 10 | $10^4$ | 0.09 ± 0.04 | 0.09 ± 0.05 | **0.51 ± 0.06** | $4.0 \cdot 10^5$ | 97.50% |
| SH | 10 | $10^5$ | 0.10 ± 0.05 | 0.09 ± 0.04 | 0.37 ± 0.03 | $7.3 \cdot 10^5$ | 86.30% |
| SH LMC | 10 | $10^5$ | 0.10 ± 0.04 | **0.07 ± 0.05** | **0.35 ± 0.05** | $7.3 \cdot 10^5$ | 86.30% |
| SH | 20 | $10^4$ | 0.12 ± 0.04 | **0.06 ± 0.04** | 0.67 ± 0.03 | $7.7 \cdot 10^5$ | 98.70% |
| SH LMC | 20 | $10^4$ | **0.11 ± 0.07** | 0.09 ± 0.08 | **0.59 ± 0.05** | $7.7 \cdot 10^5$ | 98.70% |
| SH | 20 | $10^5$ | **0.07 ± 0.01** | 0.07 ± 0.01 | 0.39 ± 0.01 | $1.4 \cdot 10^6$ | 92.86% |
| SH LMC | 20 | $10^5$ | 0.08 ± 0.03 | **0.06 ± 0.03** | **0.36 ± 0.04** | $1.4 \cdot 10^6$ | 92.86% |

under lower compute budgets, the GP receives fewer training points (due to shorter LCs) and hence exhibits wider confidence intervals, i.e., higher uncertainty and variance, which leads to higher AbC values. Conversely, higher budgets

Table 4: Area between Curves (AbC) between the ground truth SL and the predicted SL using SH LMC, extrapolation with a GP mean, GP UCB or GP LCB. Budget $B$ in petaFLOPs. Averaged over 10 runs.

| $B$ | ↓ AbC SH LMC | ↓ AbC GP Mean | ↓ AbC UCB | ↓ AbC LCB |
|-----|------|------|------|------|
| $10^3$ | 5.84 | $\mathbf{0.51 \pm 0.27}$ | $0.62 \pm 0.27$ | $0.49 \pm 0.16$ |
| $10^4$ | 3.88 | $\mathbf{0.36 \pm 0.42}$ | $0.48 \pm 0.13$ | $0.45 \pm 0.19$ |
| $10^5$ | 2.17 | $\mathbf{0.00 \pm 0.00}$ | $0.53 \pm 0.31$ | $0.38 \pm 0.16$ |

improve the GP's extrapolation ability and reduce AbC. The AbC values are computed between each predicted SL (GP Mean, UCB, or LCB) and the ground truth SL. On top of improving the accuracy of the SL, incorporating UCB and LCB predictions provides a range in which the true law is likely to lie, and hence offers a quantified measure of uncertainty that supports more informed decision-making.

## 5 CONCLUSION

We have demonstrated that Successive Halving, particularly when combined with surrogate models, enables informed and efficient resource allocation, yielding learning curves well-suited for scaling law estimation. The resulting scaling laws closely approximate the ground truth while requiring only a fraction of the compute of traditional approaches. Leveraging the LMC model for extrapolation further improves accuracy and allows estimation of the $L_{SL}(C)$ law even in regions beyond the available budget. Fitting to the GP's upper and lower confidence bounds additionally provides meaningful uncertainty estimates. Given the compute constraints faced by many researchers, we hope this work encourages a greater focus on computational efficiency in scaling law research.

## 6 REPRODUCIBILITY STATEMENT

In our work, we estimate the $L(C)$ scaling law. Further details on the $L(C)$ scaling law can be found in Appendix A and in Hoffmann et al. (2022). We use the $L(N, D)$ scaling law to generate synthetic datasets. Additional information is provided in Appendix H, and the functional forms of the noise models are given in Appendix J. We use nanoGPT to construct our real-world dataset (Karpathy, 2022). All preprocessing steps of the learning curves, including the hyperparameters used for the nanoGPT models, are reported in Appendix M, and details of the surrogate fit to the nanoGPT learning curve dataset are provided in Appendix N. We describe all surrogate models in detail in Appendix F, and we provide a visual illustration of the AbC measure in Appendix Q.

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

# A  SCALING LAWS

The recent surge in scaling law research offers several benefits, including improved interpretability of neural networks (Bahri et al., 2024; Abnar et al., 2022; Bansal et al., 2022), more effective training dataset size acquisition, and a reduced carbon footprint (Domhan et al., 2015; Johnson et al., 2018). While theoretical approaches have been developed to facilitate the prediction of scaling laws for only a limited number of architectures (Bahri et al., 2024; Hutter, 2021; Sharma & Kaplan, 2022), the majority of successful methods for predicting scaling behavior across a wide range of networks are based on empirical studies (Kaplan et al., 2020; Henighan et al., 2020; Hägele et al., 2024; Hoffmann et al., 2022; Bansal et al., 2022; Ghorbani et al., 2022; Hestness et al., 2017; Johnson et al., 2018; Rosenfeld, 2021).

Scaling laws depict a functional form $f(x)$ that characterizes the development of a performance measure, such as validation loss $L$, relative to a scale, which may refer to compute $C$, the number of samples in the dataset $D$, or the number of parameters in the model $N$. In an empirical approach, LCs are generated, and the parameters of the functional form $f(x) = \beta x^a + c$ are then estimated, for $x$ being the scale of interest, $c$ denoting the irreducible loss (Henighan et al., 2020; Hägele et al., 2024) and $f(\cdot)$ being the functional form of its performance measure Alabdulmohsin et al. (2022).

Hoffmann et al. (2022) describe three approaches to deriving scaling laws. All three approaches yield broadly comparable, though slightly different, results.

**Approach 1.** This approach fixes the model size $N$ and varies the number of training tokens $D$, yielding the $L(C)$ scaling law, $L(C) = (C/\alpha)^{-\gamma}$, where $\alpha$ and $\gamma$ are parameters to be estimated. Once the learning curves for the models of interest are obtained, the $L(C)$ law is fit to the compute-efficient frontier within the compute region under consideration.

**Approach 2.** This approach makes use of IsoFLOP profiles.

**Approach 3.** This approach models the joint loss function $L(N, D)$, which is discussed in more detail in Appendix H.

While all three methods lead to broadly similar scaling laws, they do exhibit some deviations (Hoffmann et al., 2022). In our work, we generate synthetic datasets using Approach 3, but estimate scaling laws based on Approach 1.

Scaling laws operate under the assumption of an ideal scenario where increasing dataset size corresponds to having consistently same-quality data available. However, as noted in Muennighoff et al. (2024), this is not the reality for most languages, where available data is limited, leading to data-constrained large language models (LLMs). Even for English, high-quality data is predicted to be exhausted by 2024 according to the Chinchilla scaling laws, given the current trend of training ever-larger models (Villalobos et al., 2024). In data-limited regimes, strategies such as repeating data can alter scaling law behavior. Moreover, as mentioned in Kaplan et al. (2020), scaling law trends must eventually level off, a phenomenon that a straightforward power-law trend cannot predict. In certain cases, such as discriminative models like image classifiers, clear scaling laws do not emerge, and performance may plateau even as dataset or model size increases (Hendrycks, 2024).

In a traditional setting without budget constraint, learning curves for scaling law research are typically obtained by training models until convergence (Kaplan et al., 2020; Hoffmann et al., 2022). Table 3 presents this case in the column "Cost Entire LCs", which reflects the computational cost incurred when fully training the models without any budget allocation strategy; whereas $B$ represents the budget allocated by our SH or SH LMC. Notably, this table shows that our strategic methods achieve a scaling law comparable to that obtained using the traditional approach (AbC Entire LCs), while being significantly more cost-efficient.

# B  More Details on the Proposed Algorithms

## B.1  Strategic Compute Allocation

---

**Algorithm 1** Strategic Compute Allocation

---

**Input:** $B$: Total compute budget in FLOPs.
$\mathcal{M}_{r=0}$: Set of models in round $r = 0$.
$\eta$: Pruning parameter.
**Output:** $\mathcal{L}$: Set of learning curves.
**Initialize:** Models $m \in \mathcal{M}_0$.

1: **for** $r \leftarrow 0$ to $\lceil \log_\eta |\mathcal{M}_0| \rceil - 1$ **do**
2: $\quad C_r \leftarrow \left\lfloor \frac{B}{|\mathcal{M}_r| \cdot \lceil \log_\eta |\mathcal{M}_0| \rceil} \right\rfloor$
3: $\quad \tilde{\mathcal{L}} = \{\texttt{obtain\_LCs}(m, C_{0:r}): m \in \mathcal{M}_r\}$
4: $\quad \mathcal{M}_{r+1} = \texttt{Top\_k}(\mathcal{M}_r, \tilde{\mathcal{L}}, \eta)$
5: $\quad \mathcal{L} \leftarrow$ Update by $\tilde{\mathcal{L}} \setminus \hat{\mathcal{L}}$.
6: **end for**
$\quad$ **Return:** $\mathcal{L}$.

---

The main structure of our approach is presented in Algorithm 1. The inputs are the total available compute budget $B$ (in FLOPs), the set of models $\mathcal{M}$ in the first round $r = 0$, i.e., $\mathcal{M}_{r=0} = \mathcal{M}_0$ and the pruning parameter $\eta$. The output is the set $\mathcal{L}$ after the last round $r = \lceil \log_\eta |\mathcal{M}_0| \rceil - 1$, which contains all obtained learning curves. The algorithm is initialized with the models $m$ in $\mathcal{M}_0$, where each model has a distinct number of parameters $N_m$, i.e., all models in $\mathcal{M}_0$ differ in size. Each line in Algorithm 1 can be described as follows.

1. The algorithm starts with round $r = 0$. Depending on $\eta$ and the total number of models of interest $|\mathcal{M}_0|$, the final round is reached at $r = \lceil \log_\eta |\mathcal{M}_0| \rceil - 1$ (Jamieson & Talwalkar, 2016).

2. A budget $C_r$ is allocated to each model $m$ in round $r$. This allocation depends on the current round $r$, the set of models $\mathcal{M}_r$, $\eta$, $\mathcal{M}_0$, and $B$ (Jamieson & Talwalkar, 2016).

3. All models in the current round $r$, i.e., $m \in \mathcal{M}_r$, together with the allocated budget $C_r$ and the history of previous budget allocations $C_{0:r}$, are passed to the `obtain_LCs` function (see Appendix B.2). In this step, each model is trained with budget $C_r$, either for the first time ($r = 0$) or as a continuation of its learning curve. Afterwards, the algorithm has the following options.

    a) If SH with surrogates is applied, the surrogates predict the continuation of each model's learning curve under the assumption that the model advances to the final round, i.e., up to the maximum budget that would ultimately be allocated to it. The output $\tilde{\mathcal{L}}$ is then a set of learning curves consisting of both the observed part (trained with $C_r$) and the predicted continuation.

    b) If only SH is applied, the output $\tilde{\mathcal{L}}$ consists solely of the learning curves observed after training with $C_r$.

4. The function `Top_k` selects the $k$ best models from $\mathcal{M}_r$ based on the lowest loss value their learning curves in $\tilde{\mathcal{L}}$ achieve, where $k$ is determined by the pruning parameter $\eta$. These models are passed to the next round in the set $\mathcal{M}_{r+1}$ (Jamieson & Talwalkar, 2016).

5. The set $\mathcal{L}$ collects all learning curves excluding the predicted continuations. (The latter belong to $\tilde{\mathcal{L}}$ if SH with surrogates is used and are also stored in $\hat{\mathcal{L}}$, see Algorithm B.2, line 13). Hence, we update by $\tilde{\mathcal{L}} \setminus \hat{\mathcal{L}}$. A model $m$ might receive multiple budget allocations, because it remains in the set $\mathcal{M}_{r+1}$ for consecutive rounds. Therefore, when updating $\mathcal{L}$, only the learning curve corresponding to the largest allocated compute budget is retained. In other words, existing curves in $\mathcal{L}$ are replaced by their most recent variant, which always extends furthest in training.

## B.2 Obtain LCs

---

**Algorithm 2** `obtain_LCs`$(m, C_{0:r})$: $m \in \mathcal{M}_r$

---

1: **for** $m \leftarrow 0$ to $|\mathcal{M}_r| - 1$ **do**
2: $\quad D_m^r = \frac{C_r}{6N_m}$
3: $\quad$ **if** $r = 0$ **then**
4: $\quad\quad$ Train model $m$ for one epoch using $D_m^0$ and $C_r$.
5: $\quad\quad \tilde{l} = L_m(C_r)$: Obtain learning curve for model $m$.
6: $\quad$ **else**
7: $\quad\quad C_m = \sum_{i=0}^{r} C_i$: Get the total allocated budget to model $m$ until this round $r$.
8: $\quad\quad$ Continue training model $m$ for $D_m^r$ and $C_r$.
9: $\quad\quad \tilde{l} = L_m(C_m)$: Update learning curve for model $m$.
10: $\quad$ **end if**
11: $\quad$ **if** A surrogate model is used **then**
12: $\quad\quad \hat{l} = L_m(C_{\tilde{m}})$: Predict validation loss using a surrogate model for the available budget per learning curve assuming it reaches the final round, $C_{\tilde{m}} = C_{\lceil \log_\eta |\mathcal{M}_0| \rceil - 1}$.
13: $\quad\quad \hat{\mathcal{L}} \leftarrow \hat{l}$: Add the predicted continuation of learning curve $\hat{l}$ to set $\hat{\mathcal{L}}$.
14: $\quad$ **else**
15: $\quad\quad \hat{\mathcal{L}} \leftarrow \emptyset$: No prediction is made, i.e., empty set.
16: $\quad$ **end if**
17: $\quad \tilde{\mathcal{L}} \leftarrow \tilde{l}$: Add learning curve $\tilde{l}$ to set $\tilde{\mathcal{L}}$.
18: $\quad \tilde{\mathcal{L}} \leftarrow \tilde{\mathcal{L}} \cup \hat{\mathcal{L}}$: Merge set $\tilde{\mathcal{L}}$ and $\hat{\mathcal{L}}$.
19: **end for**
$\quad$ **Return:** $\tilde{\mathcal{L}}$: A list of all obtained learning curves.

---

Algorithm 2 describes how the set of learning curves $\tilde{\mathcal{L}}$ gets created in each round $r$. In round $r$, the input to this algorithm are the models $m \in \mathcal{M}_r$ and the compute budgets allocated from the very first round $r = 0$ until the current round $r$ given by $C_{0:r}$. Afterwards, each line in Algorithm 2 can be described as follows. Note, each learning curve of a model $m$ is a curve describing the trend of the loss over the compute budget. $L_m(C_x)$ denotes a learning curve for model $m$ trained until compute budget $C_x$. In our case, $x$ stands representative for $r$, $m$ or $\tilde{m}$. These variables are explained in the following.

1. The algorithm loops over all available models $m \in \mathcal{M}_r$. Note, each model has a certain number of model parameters $N_m$, which are part of the model and do not need to be passed explicitly to the algorithm.

2. We calculate the correct number of allocated tokens $D_m^r$ for a model $m$ in round $r$ based on the formula $C_m^r = 6N_m^r D_m^r$. (For more information about how to get the correct $D_m^r$ for the real-world nanoGPT dataset, please refer to Appendix G.)

3. Distinguish whether this is the very first round, i.e., $r = 0$.

4. If yes, train your model $m$ for one epoch using the calculated $D_m^0$ from line 2. If a synthetic dataset is used, apply the formula for $L(N, D)$ as described in Appendix H.

5. $\tilde{l} = L_m(C_r)$ is the obtained learning curve for model $m$, using compute budget $C_r$.

6. If it is not the very first round, i.e., $r > 0$ do the following.

7. Calculate the sum of all allocated budgets until and including this round $C_m = \sum_{i=0}^{r} C_i$. Note, in each round $r$, a unique $C_r$ gets allocated to all models in this round as shown in Algorithm 1, line 2.

8. Each model has already been trained in the round(s) before, i.e., a learning curve exists. This model is now continued training using the $D_m^r$ and $C_r$ from this round.

9. We update the learning curve for model $m$ by $\tilde{l} = L_m(C_m)$.

10. The first if-loop is closed.

11. If we assume, that a surrogate model is to be used, i.e., SH LMC or SH DE, then the algorithm does the following.

12. The available budget per learning curve, assuming this learning curve reaches the final round, is denoted by $C_{\tilde{m}} = C_{\lceil \log_\eta |\mathcal{M}| \rceil - 1}$. This budget is used by the surrogate model to predict the continuation of the attained loss of the learning curve from the last allocated compute value $C_m$ until the total available budget $C_{\tilde{m}}$.

13. We add the predicted continuation of this learning curve $\hat{l}$ to set $\hat{\mathcal{L}}$.

14. If we assume, that a surrogate model is not used, i.e., we continue with simple SH, then the following is done.

15. We keep the learning curve as obtained in line 9 or 5 and omit any prediction. Hence, the set $\hat{\mathcal{L}}$ is empty.

16. The second if-loop is closed.

17. Add each obtained learning curve after training to a list $\tilde{\mathcal{L}}$

18. Merge the trained and predicted learning curves for a model, contained in set $\tilde{\mathcal{L}}$ and $\hat{\mathcal{L}}$, respectively. Keep the final learning curve in set $\tilde{\mathcal{L}}$

19. The for-loop is closed.

Algorithm 2 returns a list $\tilde{\mathcal{L}}$ of all obtained learning curves $\tilde{l}$ for all models $m$ in the current round $r$. If SH with surrogates was used, each learning curve will additionally contain the predicted continuations $\hat{l}$.

## C   MORE INFORMATION ABOUT SH VS. SH+SURROGATE AND HOW TO READ LEARNING CURVE FIGURES

Due to the nature of our problem, the SH algorithm (Jamieson & Talwalkar, 2016) in its original form without integrated surrogate prediction would inherently favor smaller models in each round $r$ for smaller $C_m^r$ values: Small models generally need less compute to fulfill one training iteration and reach smaller loss values earlier (see Figure C.1). However, they will also plateau earlier and are therefore naturally not able to achieve a loss value as small as a bigger model could potentially achieve later on – rendering this naive form of the SH algorithm suboptimal for our scenario.

A side-by-side illustration of the budget allocation approach using SH and SH LMC (one of our surrogates) is shown in Figure C.1. In each round $r$ the same compute is allocated to all models, and the LCs therefore stop at the same compute value. As previously detailed, only the 'most-promising' models selected to train further will be allocated more compute in the next round – resulting in the

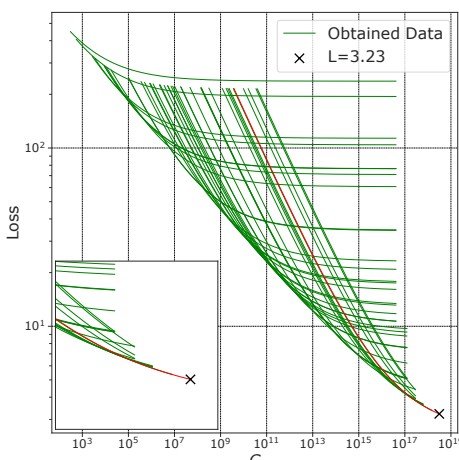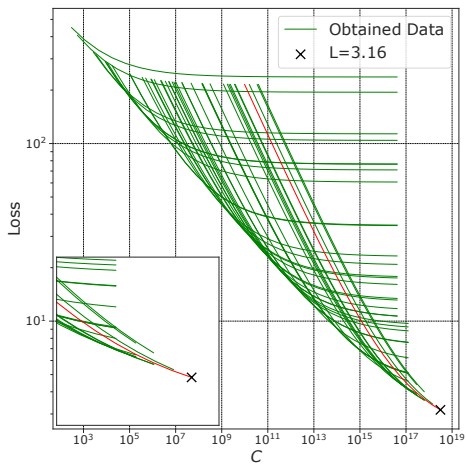

Figure C.1: Budget allocation on a synthetic LC dataset. **Left**: Minimum test loss obtained for SH. **Right**: Minimum test loss obtained for our surrogate-integrated approach SH LMC.

different end-compute values for the LCs (here between $10^{16}$-$10^{19}$), based on the number of rounds the models participated in. Note that predicting the likely future development of an LC while taking both the model $m$ and its maximally remaining budget $C_{\tilde{m}}$ into account leads to a different model selection as shown in Figure C.1 (highlighted in red), which eventually reaches a lower loss value. In other words, combining SH with surrogate models to predict a model's *future* performance is essential for making informed and strategic decisions about budget allocation and identifying which models warrant further training, particularly when their *current* performance might not yet provide a clear indication of their true potential.

## D  AN EXAMPLE FOR SH LMC USING 20 MODELS.

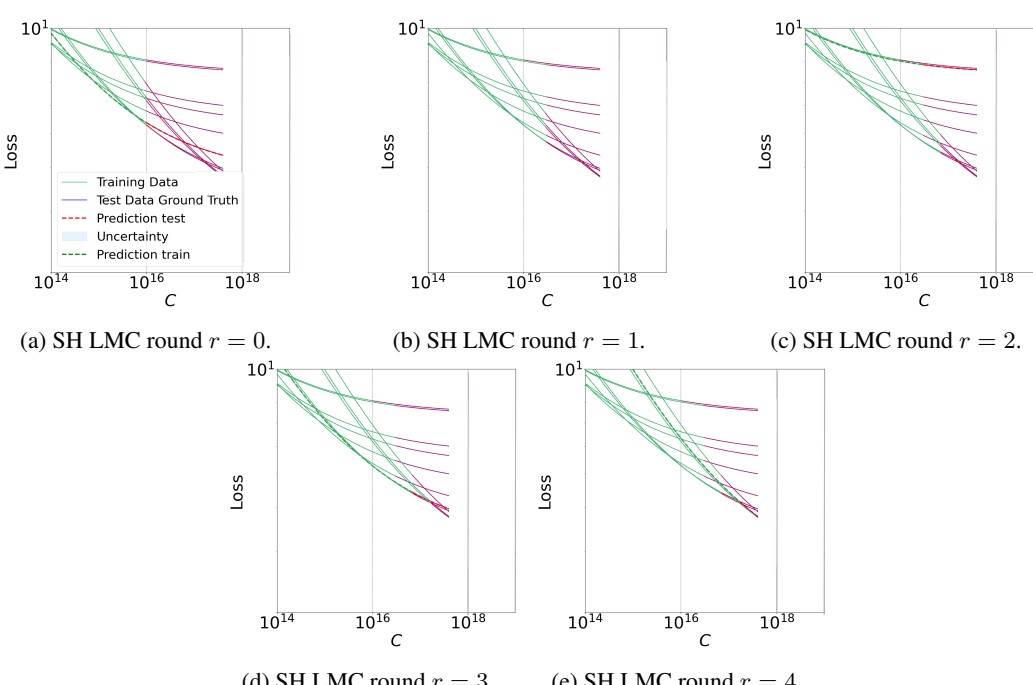

(a) SH LMC round $r = 0$.   (b) SH LMC round $r = 1$.   (c) SH LMC round $r = 2$.

(d) SH LMC round $r = 3$.   (e) SH LMC round $r = 4$.

Figure D.1: An example of how SH LMC allocates budget to models in 5 rounds. Ground truth training data shown in green lines, predicted training data shown in dashed green lines, ground truth test data shown in blue and predicted test data shown in dashed red lines.

Figure D.1 illustrates how SH LMC allocates budget to models based on the information in their learning curves. Each subfigure shows the learning curves of models $m \in \mathcal{M}_0$ at different rounds of the SH LMC algorithm. The models $m$ vary in size. Smaller models tend to plateau earlier in their learning curves. Solid green lines show the learning curves of trained models. Additionally, solid green lines are also the surrogate model's training data, while dashed green lines show the corresponding fit. The ground truth for the test data is shown in blue, and the surrogate's predictions are shown as red dashed lines. Prediction uncertainty is highlighted in light blue. Note that, since a synthetic dataset is used for this toy example, models may be of similar size, resulting in learning curves that appear very close to each other in the subfigures. Also, without any loss of generality, but to avoid unnecessary cluttering, all subfigures are zoomed onto the best 12 learning curves. SH LMC progresses as described in the following. (Please also refer to Appendix B.)

    a) In the initial round ($r = 0$), all models $m \in \mathcal{M}_0$ are allocated the same compute budget $C_r$. Accordingly, each model is trained until its learning curve corresponds to the green solid lines. These are stored in $\tilde{\mathcal{L}}$. Although the budget is identical across models, their differing sizes result in learning curves that reach different loss values. SH LMC uses these learning curves as training data to predict their continuation, shown by the red dashed lines. This part is stored in the set $\hat{\mathcal{L}}$. Later, in `Top_k`, the predicted minimum loss values (from the

red dashed lines) are used to determine which models are retained for the next round of the algorithm. With $\eta = 2$, the best 10 models are retained for the next round, as shown in b). These are the models whose predicted loss values are smallest relative to the full set. (Note, if only SH is used, `Top_k` uses the minimum loss values of the obtained learning curves, i.e., the green solid lines to obtain the best model for the next round.)

b) In round $r = 1$, only 10 models are left, these are allocated the same budget as in the round before. Note that in this subfigure the smallest model is not further trained; only the best 10 models continue their learning curves. Based on their predicted continuations, 5 models are retained for the next round. Note that predictions are made only for the 10 selected models; however, for demonstration purposes, the subfigures display the predicted continuations for all learning curves.

c) In round $r = 2$, the learning curves of the continued models show that the algorithm correctly retained the larger models, as they achieve lower predicted loss values. This is visualized by the green solid lines.

d) In round $r = 3$, the algorithm retains 2 models. As shown in the subfigure, one of them indeed has the potential to achieve the lowest loss value among all models in the set.

e) Finally, in the last round, additional budget is allocated to the model achieving the lowest loss value. Throughout this process, all obtained learning curves (green solid lines) along with their predictions (red dashed lines) were stored in the set $\tilde{\mathcal{L}}$. The output of the SH LMC algorithm is the set of all obtained learning curves (green solid lines) without their predictions (red dashed lines; see Appendix B.1, line 5).

Note the prediction quality of the LMC surrogate model, which aligns well with the underlying ground truth values.

## E EXAMPLES COMPARING SH VS. SH LMC

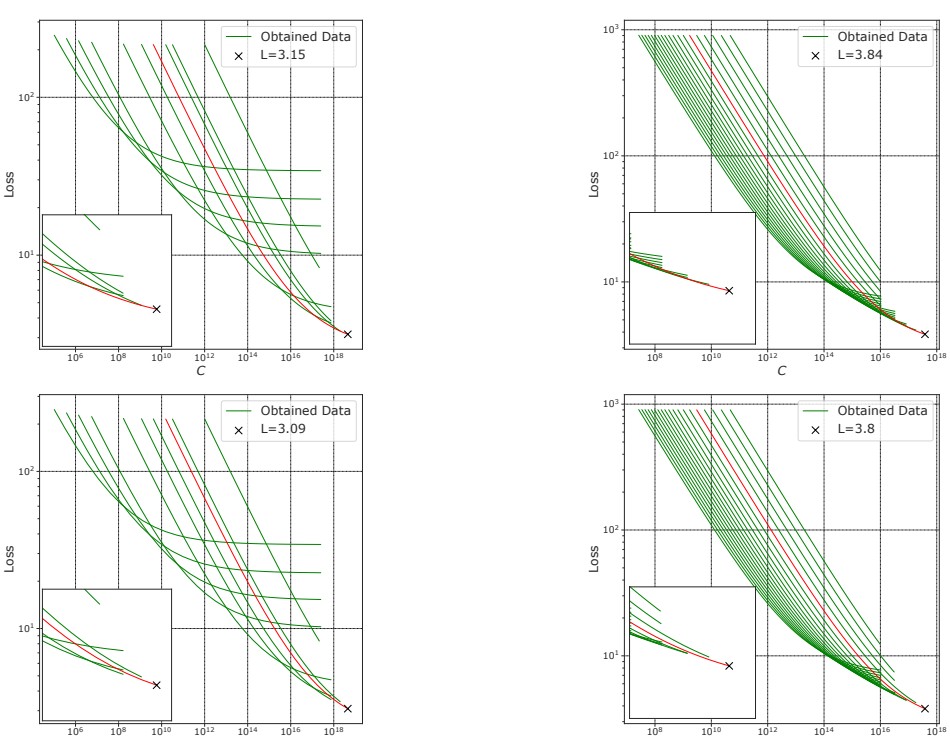

Figure E.1: For two different datasets: **Top**: Minimum test loss obtained when allocating compute budget according to SH. **Bottom**: Minimum test loss obtained when allocating compute budget according to SH LMC.

An example for budget allocation according to SH and SH LMC is given in Figure E.1. On the left the comparison using 10 LCs is given. SH LMC allocates more budget to a larger model, which leads to a lower loss value being 3.09. In comparison SH allocates the budget such that the lowest loss value obtained is 3.15. On the right, the second synthetic dataset from (Pearce & Song, 2024) was used and similar results can be shown. While SH achieves a loss value of 3.84, SH LMC achieves 3.80.

# F   FURTHER DETAILS ON THE SURROGATE MODELS

## F.1   THE LMC MODEL

The LMC assumes that the output $\mathbf{y} \in \mathbb{R}^N$ of a LC $i$ can be modeled as a linear combination of $l$ independent Gaussian Processes, for $i = 1, ..., Q$, given as

$$\mathbf{y}_i = \mathbf{W}\mathbf{f}_i + \epsilon \quad \text{and} \quad f_{il} = f_l(\mathbf{x}_i) \quad \text{with } f_l \sim GP(\mathbf{0}, \mathbf{K}_l) \quad \text{and } \epsilon \sim N(\mathbf{0}, \sigma^2\mathbf{I}_Q),$$

where $\mathbf{f}_i$ is a vector of $L$ latent function evaluations for sample $i$, and $\mathbf{W}$ is a matrix mapping from the latent function values to the observed data outputs. $\mathbf{K}$ is the full kernel matrix and $\mathbf{K}_l$ is the $l_{\text{th}}$ kernel function evaluated at all pair of points. As all assumptions are Gaussian, the likelihood of all observations $\mathbf{Y} \in \mathbb{R}^{Q \times N}$, for $Q$ outputs and $N$ being the length of each output, is given as $\mathbf{Y} \sim \mathcal{N}(\mathbf{0}, \Sigma + \sigma^2\mathbf{I}_{NQ})$. Assuming all outputs share the same $\Sigma$, it holds that

$$\Sigma = \mathbf{K} \otimes \mathbf{B}^T, \quad \text{where } \mathbf{B} = \mathbf{W}^T\mathbf{W} + \text{diag}(\kappa).$$

The input covariance matrix is denoted by $\mathbf{K}$ and the output covariance matrix is $\mathbf{B}$. The LMC model we employ uses a composite kernel given by

$$\Sigma = \mathbf{B}_1 \otimes \mathbf{K}_{\text{ExpDec}} + \mathbf{B}_2 \otimes \mathbf{K}_{\text{White}} + \mathbf{B}_3 \otimes \mathbf{K}_{\text{Bias}}, \quad \text{with } \mathbf{B}_j = \mathbf{W}_j^T\mathbf{W}_j + \text{diag}(\kappa_j).$$

While $\mathbf{K}$ denotes the full kernel matrix, the individual kernels are defined by

$$k(x, x')_{\text{ExpDec}} = \sigma^2 \frac{\beta^\alpha}{(x + x' + \beta)^\alpha}, \quad k(x, x')_{\text{White}} = \sigma^2 I_n, \quad k(x, x')_{\text{Bias}} = \sigma_0^2,$$

where $\alpha$ and $\beta$ are hyperparameter to optimize, $\sigma^2$ denotes the variance of the kernel, $\sigma_0^2$ is a constant bias to be learnt and $I_n$ denotes the identity matrix.

**Application onto our Learning Curve Prediction Task**

The outputs of a Gaussian Process are represented as a joint multivariate Gaussian distribution, with dependencies captured through the covariance (kernel) function. Unlike models that assume independence, GPs explicitly account for correlations between outputs through the chosen kernel.

The LMC model builds on multitask Gaussian Processes, where each learning curve is treated as a separate task. This formulation enables the model to identify shared trends across tasks in the training data and to exploit these correlations for improved extrapolation.

## F.2   DEEP ENSEMBLE METHODS

The deep ensemble methods approximate the following functions

$$f_{\text{PL}}(x) = ax^{-b} + c, \quad f_{\text{EXP}}(x) = a\exp(-bx) + c,$$
$$f_{\text{MMF}}(x) = (ab + cx^d)/(b + x^d).$$

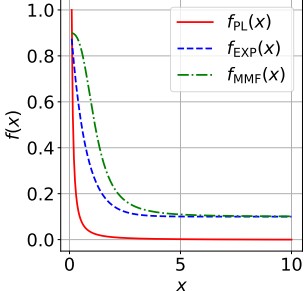

An example of the shape of these functions is given in Figure F.1. We define the learning task of the Deep Neural Network (DNN) according to Kadra et al. (2023) as learning the coefficients of the equations defined above, conditioned on the model sizes. In our scenario the learning curves differ because they are obtained after training models of different sizes. Therefore, the input to the DNN is the model size and the output will be the obtained loss value, for a set of learnt coefficients. The network itself is a fully-connected two hidden layer neural network.

Figure F.1: An example of the power law, the exponential and the MMF function.

## G  CALCULATION OF $D_m$

We assume the approximation $C_m = 6N_m D_m$ (Kaplan et al., 2020). In a real-world experiment we have

$$D_m = \text{steps}_m \cdot B \cdot T \cdot \text{grad-accum} \cdot \text{ddp-world-size},$$

where $B$ denotes batchsize, $T$ is the sequence length, grad-accum are the gradient accumulation steps and ddp-world-size the number of GPUs used. Assume

$$x = B \cdot T \cdot \text{grad-accum} \cdot \text{ddp-world-size}.$$

Then

$$\text{steps}_m = \left\lfloor \frac{C_m}{N_m x} \right\rfloor, \tag{3}$$

giving us the correct steps size to train our model and

$$\tilde{D}_m = \text{steps}_m x.$$

In our SH approach, we assume a step size calculation for both the synthetic and nanoGPT datasets as outlined earlier. This method employs a rounding-down operation to allocate the budget and determine step sizes, which can leave some budget unused. This remaining budget can be leveraged in subsequent rounds to further reduce test loss in both SH and SH plus surrogate methods.

## H  SYNTHETIC DATA SET BASED ON $L(N, D)$

Table H.1: Parameters of the $L(N, D)$ loss function used in our study.

|  | $N_c$ | $D_c$ | $E$ | $\alpha$ | $\beta$ |
|---|---|---|---|---|---|
| Besiroglu et al. (2024) | 482.01 | 2085.43 | 1.8172 | 0.3392 | 0.2849 |
| Hoffmann et al. (2022) | 406.40 | 410.7 | 1.6934 | 0.3478 | 0.3658 |

An alternative scaling law is the $L(N, D)$ scaling law, which is obtained by fitting the following equation to the $N$, $D$ and final validation loss values of the empirically obtained LCs (Pearce & Song, 2024; Hoffmann et al., 2022)

$$L(N, D) = \frac{N_c}{N^\alpha} + \frac{D_c}{D^\beta} + E, \tag{4}$$

with $\{N_c, D_c, E, \alpha, \beta\}$ being the set of parameters to be optimized. In order to obtain the parameters the Huber loss is minimized between the predicted and observed loss using the L-BFGS algorithm (Hoffmann et al., 2022)

$$\min_{A, B, E, \alpha, \beta} \sum_{\text{runs i}} \text{Huber}_\delta \left( \log \hat{L}(N_i, D_i) - \log L_i \right),$$

with $\delta = 10^{-3}$. The optimal parameters according to Hoffmann et al. (2022) and Besiroglu et al. (2024) are given in Table H.1.

As stated in Hoffmann et al. (2022), $E$ corresponds to the entropy of natural text and captures the irreducible error of the model. The irreducible loss does not disappear, even with infinite model size and infinite data. A linear scaling law is only an approximation valid far above that floor; as performance approaches the irreducible term, improvement necessarily slows and the curve flattens.

For our experiments we build on and extend the library *Chinchilla*[1].

## I  SCALING LAWS FOR VARYING MODEL SETS

Figure I.1 shows how the model sizes in the set of models of interest affect the scaling law if the compute efficient frontier is obtained in the same compute range. The left figure shows a scaling law obtained for large models, while the right figures uses smaller models.

---

[1]`https://github.com/kyo-takano/chinchilla/tree/master`

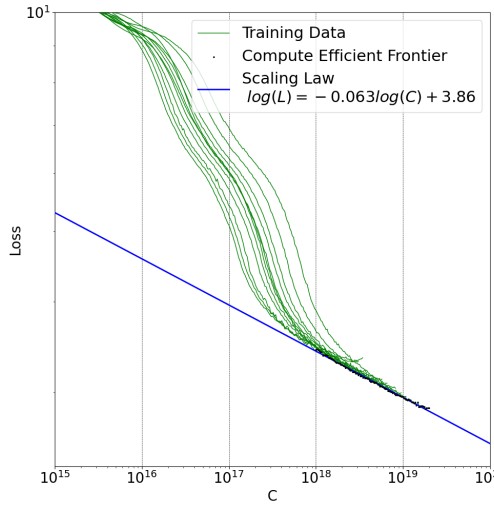 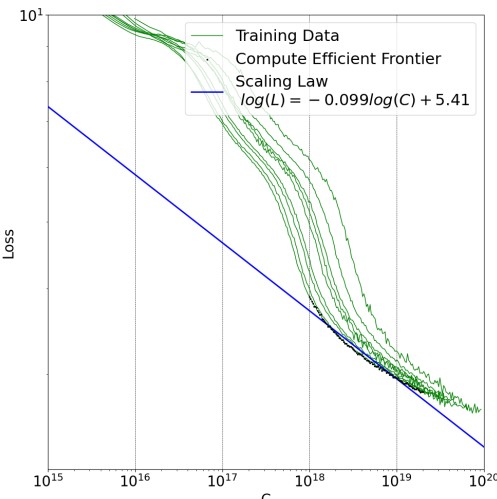

(a) The obtained scaling law using LCs of a set of models consisting of smaller models.

(b) The obtained scaling law using LCs of a set of models consisting of larger models.

Figure I.1: Scaling Laws for two different sets of models.

## J  NOISE MODELS

For all noise models, we assume the intensity is weighted from $w \in [0, ..., 1]$ applied until either the inclination point of the LC is reached, which we measure using the gradient of the curve, or the end of the available training data points. Note, noise is only added to the synthetic dataset, i.e., this is feasible in this case. The noise $n$ is added to the log loss as

$$\log(L)_{\text{noisy}} = L(N, D) + n \tag{5}$$

**AWGN.**  To add Gaussian noise, we apply the weight $w$ to the noise samples, sampled from $n_{\text{AWGN}} \sim \mathcal{N}(0, w\sigma)$, with $\sigma$ being the standard deviation.

**Browninan Noise.**  Brownian noise is simulated by

$$n_{\text{Brown}}(t + \mathrm{d}t) = n_{\text{Brown}}(t) + \mathcal{N}(0, w\sigma^2 \mathrm{d}t),$$

with $\mathrm{d}t$ being the compute-step.

**Ornstein-Uhlenbeck Noise.**  Ornstein-Uhlenbeck noise is simulated by

$$n_{\text{OU}}(t+h) = \mu + (n_{\text{OU}}(t) - \mu)\exp(-h/\tau) + \sigma\sqrt{(1 - \exp(-2h/\tau))} \cdot \mathcal{N}(0, 1), \quad \text{with } n_{\text{OU}} = w n_{\text{OU}}$$

with $\mu$ being the noise value, time constant $\tau$, and $h$ being the interval length.

Examples of SH LMC applied to noisy LCs are given in Figure J.1.

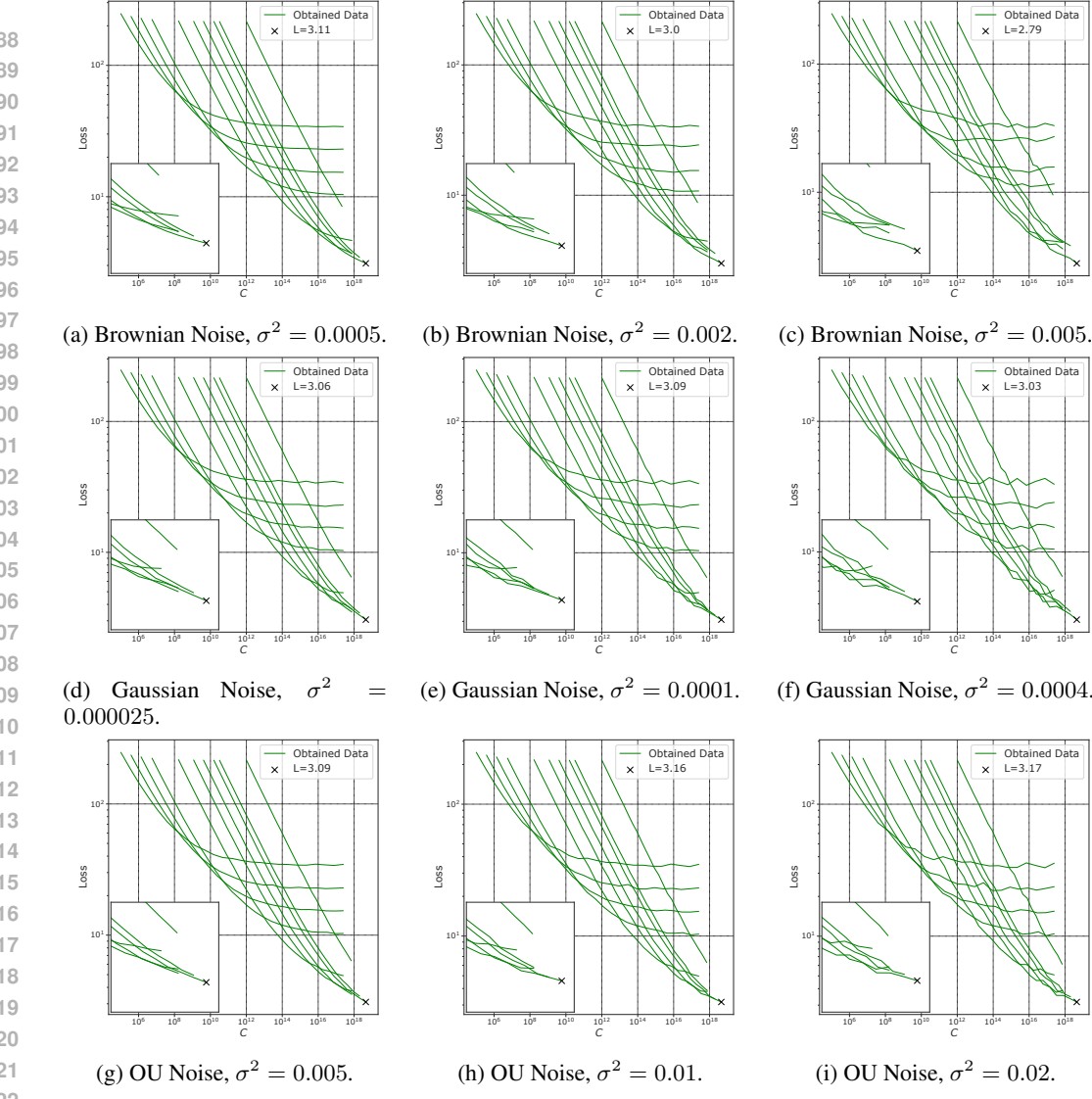

(a) Brownian Noise, $\sigma^2 = 0.0005$.  (b) Brownian Noise, $\sigma^2 = 0.002$.  (c) Brownian Noise, $\sigma^2 = 0.005$.

(d) Gaussian Noise, $\sigma^2 = 0.000025$.  (e) Gaussian Noise, $\sigma^2 = 0.0001$.  (f) Gaussian Noise, $\sigma^2 = 0.0004$.

(g) OU Noise, $\sigma^2 = 0.005$.  (h) OU Noise, $\sigma^2 = 0.01$.  (i) OU Noise, $\sigma^2 = 0.02$.

Figure J.1: Noisy LCs for various values of noise intensity $\sigma^2$ and noise models.

## K FURTHER ANALYSIS USING SH WITH SURROGATE MODELS (DATASET: SYNTHETIC)

Table K.1: Mean degradation of SH LMC. The average is taken out of 100 runs. In each run models were selected randomly. The models whose learning curves are used here are part of the synthetic dataset.

| $M_0$ | $B$ | SH mean loss | LMC mean relative degradation |
|---|---|---|---|
| 5 | $10^2$ | $6.40 \pm 9.07$ | $0.00\%_{(0.00\%)}$ |
| 5 | $10^3$ | $4.62 \pm 3.10$ | $-0.07\%_{(-0.07\%)}$ |
| 5 | $10^4$ | $3.84 \pm 2.03$ | $0.00\%_{(0.00\%)}$ |
| 10 | $10^2$ | $4.73 \pm 0.49$ | $-1.37\%_{(-2.29\%)}$ |
| 10 | $10^3$ | $3.86 \pm 0.38$ | $-1.44\%_{(-1.95\%)}$ |
| 10 | $10^4$ | $3.26 \pm 0.29$ | $-2.34\%_{(-2.34\%)}$ |
| 20 | $10^2$ | $4.69 \pm 0.10$ | $0.00\%_{(0.00\%)}$ |
| 20 | $10^3$ | $3.80 \pm 0.09$ | $0.00\%_{(0.00\%)}$ |
| 20 | $10^4$ | $3.18 \pm 0.09$ | $0.00\%_{(0.00\%)}$ |

In the majority of cases SH LMC is either allocating equally well to SH or better, i.e., allocating more budget to a larger model leading to a lower loss value. However, in very few cases it performs worse. The mean relative degradation is shown in Table K.1. Notice, the mean relative degradation is much smaller than the mean relative improvement shown in Table 2. We additionally provide the average mean loss values in Table K.2.

Figure K.1 illustrates two corner cases where SH LMC performs worse than SH. These cases occur when LCs are closely placed in space.

Table K.3 provides the number of times SH LMC outperforms SH or performs equally well. On the synthetic dataset, using SH LMC can yield to performance gains. Notice, du to the nature of the problem, SH cannot be outperformed in every case. When the optimal number of models is selected by SH, SH LMC will likely make the same selection.

Table K.2: Illustration of the mean loss values for Successive Halving (SH) and SH LMC. The average is taken over 100 runs. In each run models were selected randomly. The models whose learning curves are used here are part of the synthetic dataset.

| $M_0$ | $B$ | SH mean loss | LMC mean loss |
|---|---|---|---|
| 5 | $10^2$ | $6.40 \pm 9.07$ | $\mathbf{6.37 \pm 9.07}$ |
| 5 | $10^3$ | $4.62 \pm 3.10$ | $\mathbf{4.60 \pm 3.10}$ |
| 5 | $10^4$ | $3.84 \pm 2.03$ | $\mathbf{3.80 \pm 2.01}$ |
| 10 | $10^2$ | $4.73 \pm 0.49$ | $\mathbf{4.69 \pm 0.45}$ |
| 10 | $10^3$ | $3.86 \pm 0.38$ | $\mathbf{3.85 \pm 0.38}$ |
| 10 | $10^4$ | $3.26 \pm 0.29$ | $\mathbf{3.23 \pm 0.27}$ |
| 20 | $10^2$ | $4.69 \pm 0.10$ | $\mathbf{4.66 \pm 0.07}$ |
| 20 | $10^3$ | $3.80 \pm 0.09$ | $\mathbf{3.78 \pm 0.08}$ |
| 20 | $10^4$ | $3.18 \pm 0.09$ | $\mathbf{3.17 \pm 0.07}$ |

Table K.3: Number of runs where Successive Halving (SH) is either outperformed (win) by SH LMC or shows equal performance. In each run models were selected randomly. The models whose learning curves are used here are part of the synthetic dataset.

| $M_0$ | $B$ | LMC win | equal |
|---|---|---|---|
| 5 | $10^2$ | 10 | 90 |
| 5 | $10^3$ | 10 | 89 |
| 5 | $10^4$ | 14 | 86 |
| 10 | $10^2$ | 19 | 76 |
| 10 | $10^3$ | 19 | 78 |
| 10 | $10^4$ | 18 | 81 |
| 20 | $10^2$ | 40 | 60 |
| 20 | $10^3$ | 41 | 59 |
| 20 | $10^4$ | 32 | 68 |

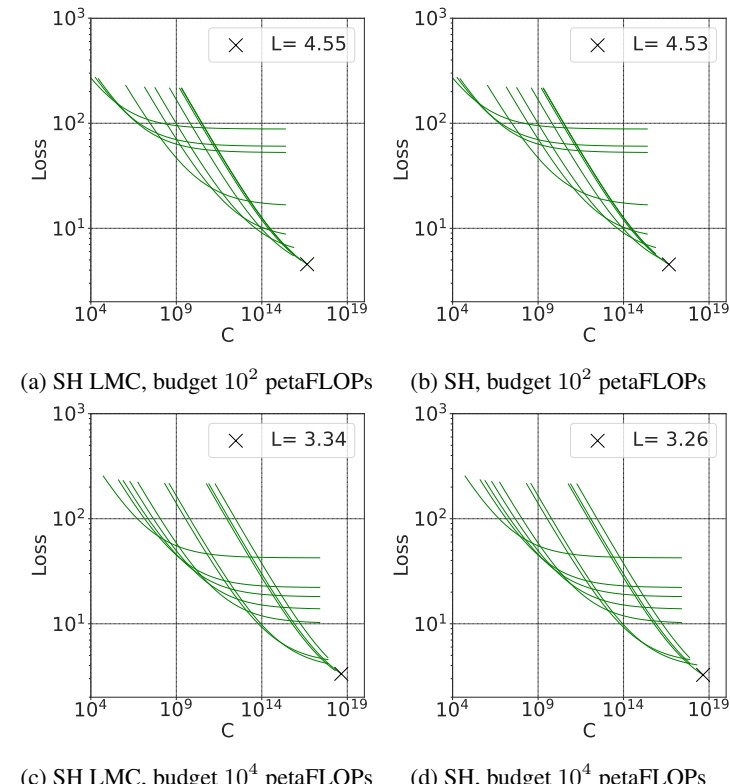

(a) SH LMC, budget $10^2$ petaFLOPs   (b) SH, budget $10^2$ petaFLOPs

(c) SH LMC, budget $10^4$ petaFLOPs   (d) SH, budget $10^4$ petaFLOPs

Figure K.1: Corner cases where SH LMC performs worse than SH assuming 10 models and different total available budget.

## L    FURTHER ANALYSIS USING SH WITH SURROGATE MODELS (DATASET: NANOGPT)

Table L.1 shows the mean relative degradation in percentage. Table L.2 reports the mean loss values, showing that on average, the lowest mean loss values are achieved by SH LMC. Table L.3 shows how often, out of 100 runs, SH in combination with a surrogate model either outperforms SH or reaches the same loss value. While the mean relative degradation is not always the smallest for SH LMC, averaged over all runs, the times it outperforms or performs equally lead to a lower mean average loss value.

Notice, the nanoGPT dataset has LCs closely placed in space (see Figure M.1), a scenario chosen similar to Kaplan et al. (2020, Figure 1, left).

Table L.1: Mean degradation of SH LMC and SH with Deep Ensembles approximating the power law (PL), exponential (EXP) and MMF function compared to Successive Halving (SH). The average is taken over 100 runs. In each run models were selected randomly using the nanoGPT dataset.

| | | SH | LMC | DE PL | DE EXP | DE MMF |
|---|---|---|---|---|---|---|
| $M_0$ | $B$ | mean loss | mean relative degradation | mean relative degradation | mean relative degradation | mean relative degradation |
| 5 | $10^4$ | $3.17 \pm 0.06$ | $-2.36\%$ $(-7.57\%)$ | $-2.32\%$ $(-42.61\%)$ | $\mathbf{-1.53}\%$ $(-5.14\%)$ | $-1.90\%$ $(-7.79\%)$ |
| 5 | $10^5$ | $2.97 \pm 0.03$ | $-0.96\%$ $(-1.80\%)$ | $\mathbf{-0.69}\%$ $(-1.83\%)$ | $-0.77\%$ $(-1.83\%)$ | $-0.88\%$ $(-1.83\%)$ |
| 10 | $10^4$ | $3.23 \pm 0.05$ | $-1.63\%$ $(-4.18\%)$ | $\mathbf{-1.26}\%$ $(-4.16\%)$ | $-1.91\%$ $(-5.56\%)$ | $-1.67\%$ $(-5.47\%)$ |
| 10 | $10^5$ | $3.00 \pm 0.02$ | $-1.59\%$ $(-4.89\%)$ | $-1.20\%$ $(-8.75\%)$ | $\mathbf{-0.86}\%$ $(-1.6\%)$ | $-4.39\%$ $(-45.8\%)$ |
| 20 | $10^4$ | $3.30 \pm 0.02$ | $\mathbf{-0.50}\%$ $(-1.33\%)$ | $-2.01\%$ $(-7.04\%)$ | $-0.98\%$ $(-2.65\%)$ | $-1.10\%$ $(-2.65\%)$ |
| 20 | $10^5$ | $3.03 \pm 0.01$ | $\mathbf{-0.84}\%$ $(-2.28\%)$ | $-1.20\%$ $(-6.52\%)$ | $-1.41\%$ $(-2.63\%)$ | $-1.96\%$ $(-6.26\%)$ |

Table L.2: Mean loss values of Successive Halving (SH) to SH LMC and SH with Deep Ensembles approximating the power law (PL), exponential (EXP) and MMF function. The average is taken over 100 runs. In each run models were selected randomly using the nanoGPT dataset.

| | | SH | LMC | DE PL | DE EXP | DE MMF |
|---|---|---|---|---|---|---|
| $M_0$ | $B$ | mean loss | mean loss | mean loss | mean loss | mean loss |
| 5 | $10^4$ | $3.17 \pm 0.06$ | $\mathbf{3.14 \pm 0.06}$ | $3.19 \pm 0.15$ | $3.19 \pm 0.04$ | $3.20 \pm 0.06$ |
| 5 | $10^5$ | $2.97 \pm 0.03$ | $\mathbf{2.91 \pm 0.05}$ | $\mathbf{2.91 \pm 0.05}$ | $2.92 \pm 0.05$ | $2.92 \pm 0.05$ |
| 10 | $10^4$ | $3.23 \pm 0.05$ | $\mathbf{3.19 \pm 0.06}$ | $3.20 \pm 0.05$ | $3.27 \pm 0.06$ | $3.23 \pm 0.05$ |
| 10 | $10^5$ | $3.00 \pm 0.02$ | $\mathbf{2.94 \pm 0.05}$ | $2.97 \pm 0.05$ | $2.95 \pm 0.03$ | $2.97 \pm 0.16$ |
| 20 | $10^4$ | $3.30 \pm 0.02$ | $\mathbf{3.23 \pm 0.05}$ | $3.25 \pm 0.04$ | $3.33 \pm 0.03$ | $3.28 \pm 0.05$ |
| 20 | $10^5$ | $3.03 \pm 0.01$ | $\mathbf{2.99 \pm 0.05}$ | $3.01 \pm 0.04$ | $3.06 \pm 0.02$ | $3.01 \pm 0.05$ |

Table L.3: Number of runs where Successive Halving (SH) is either outperformed (win) by SH LMC, SH with Deep Ensembles approximating the power law (PL), exponential (EXP) and MMF function; or they show equal performance. The average is taken over 100 runs. In each run models were selected randomly using the nanoGPT dataset.

| $M_0$ | $B$ | LMC | | DE PL | | DE EXP | | DE MMF | |
|---|---|---|---|---|---|---|---|---|---|
| | | wins | equal | wins | equal | wins | equal | wins | equal |
| 5 | $10^4$ | **46** | 32 | 30 | 8 | 21 | 3 | 20 | 5 |
| 5 | $10^5$ | 82 | 14 | **85** | 6 | 80 | 7 | 80 | 6 |
| 10 | $10^4$ | **62** | 20 | 60 | 12 | 15 | 5 | 39 | 8 |
| 10 | $10^5$ | **75** | 14 | 69 | 5 | 89 | 3 | 79 | 2 |
| 20 | $10^4$ | 78 | 15 | **88** | 4 | 14 | 2 | 60 | 7 |
| 20 | $10^5$ | 61 | 21 | 68 | 16 | 4 | 6 | **72** | 3 |

# M  HYPERPARAMETERS OF THE NANOGPT MODELS AND DATA PREPROCESSING FOR THE NANOGPT DATASET

## M.1  HYPERPARMETERS

Table M.1 presents the key hyperparameters used when training nanoGPT models on the FineWeb-edu10B dataset (Penedo et al., 2024).

| Hyperparameter | Default / Typical Value |
|---|---|
| vocab size | 50,304 (GPT-2 BPE) |
| sequence length | 1024 |
| lr scheduler | cosine decay |
| learning rate (max) | $6 \times 10^{-4}$ |
| learning rate (min) | $6 \times 10^{-5}$ |
| warmup schedule | linear |
| warmup iters | 715 (training for 10B tokens) |
| batch size | 524,288 |
| weight decay | 0.0 (none) |
| optimizer | AdamW |
| $\beta_1$ | 0.9 |
| $\beta_2$ | 0.95 |
| dropout | 0.0 (none) |
| grad clip | 1.0 |
| training precision | amp with bfloat16 |

Table M.1: Key hyperparameters used for nanoGPT training.

## M.2 DATA PREPROCESSING

Our dataset comprises 90 LCs. These were obtained for nanoGPT models with a number of layers being in $[8, 10, 12, 14, 18, 20, 24, 26, 28, 30, 32, 34, 36, 38, 40, 42, 44, 46, 48]$ and the number of embedding parameters being in $[8, 9, 10, 11, 12, 13, 14, 15, 16, 17, 18, 20, 25] \cdot 64$. The model parameter space $N$ is spanned by a combination of both and the calculation of the total model parameters follows the approach in Karpathy (2022)[2]. For the training of our nanoGPT models, we follow most of the default hyperparameters proposed by Karpathy (2022), with specific hyperparameters reported in Table M.1. To train the proposed surrogate models, we use 20 data-points per loss-compute curve.

We apply zero-one normalization of each learning curve to the minimum compute used in the current SH run and the maximum compute possible in this run, for the given budget. Furthermore, we employ a zero-one normalization for the achievable loss to the range $[0, 10]$. In this way, we ensure that the characteristic LC shape is preserved and can be approximated by the surrogate models. (Using this normalization we follow Klein et al. (2016).)

During training, the validation loss per training step is tracked, allowing us to have a large number of datapoints to experiment with. However, this makes the obtained dataset quite noisy. Therefore, we apply the Savitzky-Golay (SavGol) filter (Savitzky & Golay, 1964), which is known to maintain the original curve shape after filtering quite closely. In Figure M.2 a)-c) different window lengths are applied during filtering. While a larger window length tends to smoothen the curve very beneficially for further extrapolation models, we opt for the window length of $n = 11$ to preserve the original shape as much as possible. Furthermore, a polynomial order of 3 was applied as it gives the algorithm enough degrees of freedom to smooth but also preserve the curve shape and it has the capacity to model irregular learning curve shapes. Figure M.2 d) and e) contrasts the original set of five LCs to the final set after filtering and zero-one normalization. When working with the real-world nanoGPT dataset, e) is a representative for the curve shapes we will be working with.

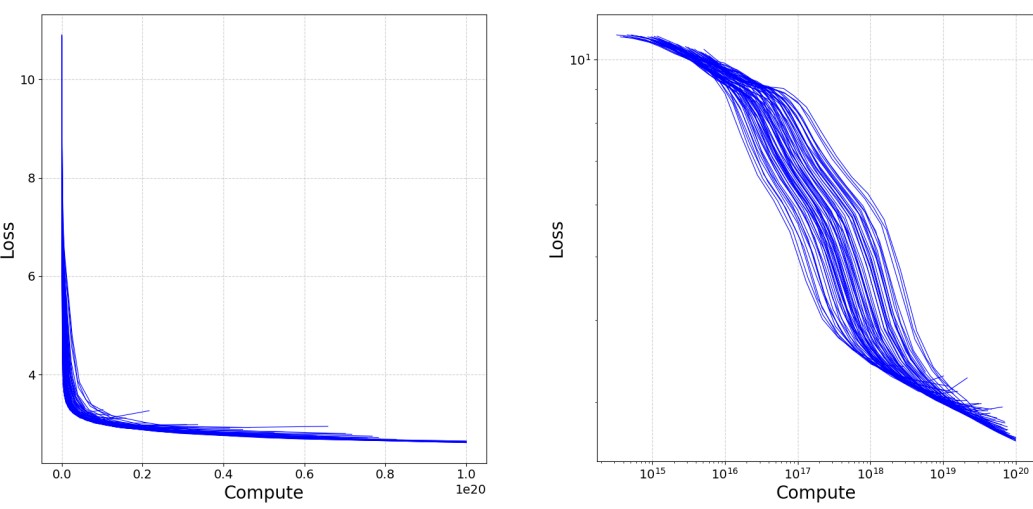

(a) The nanoGPT dataset in original domain.  (b) The nanoGPT dataset in log-log-domain.

Figure M.1: Illustration of the nanoGPT dataset.

[2]https://github.com/karpathy/nanoGPT/blob/master/scaling_laws.ipynb

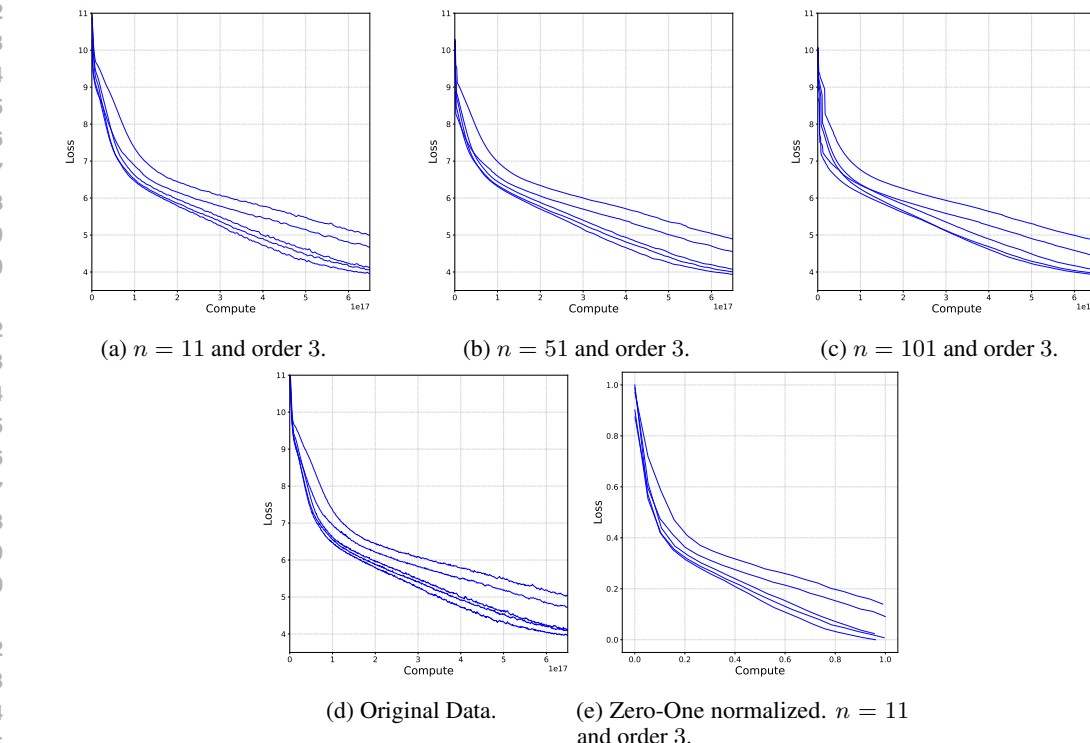

(a) $n = 11$ and order 3.     (b) $n = 51$ and order 3.     (c) $n = 101$ and order 3.

(d) Original Data.     (e) Zero-One normalized. $n = 11$ and order 3.

Figure M.2: Illustration of the data preprocessing steps using a SavGol Filter of window length $n$.

## N  SURROGATE FIT TO THE NANOGPT DATASET

The fit of the proposed surrogate models to various samples of the nanoGPT data set is shown in Figure N.1. As a preprocessing step, the loss and compute values are normalized using zero-one normalization for all surrogate models. All surrogate models manage to approximate the learning curve shape. While the multitask Gaussian Process model fits the training data tightly, the deep ensemble methods find only an approximation of it.

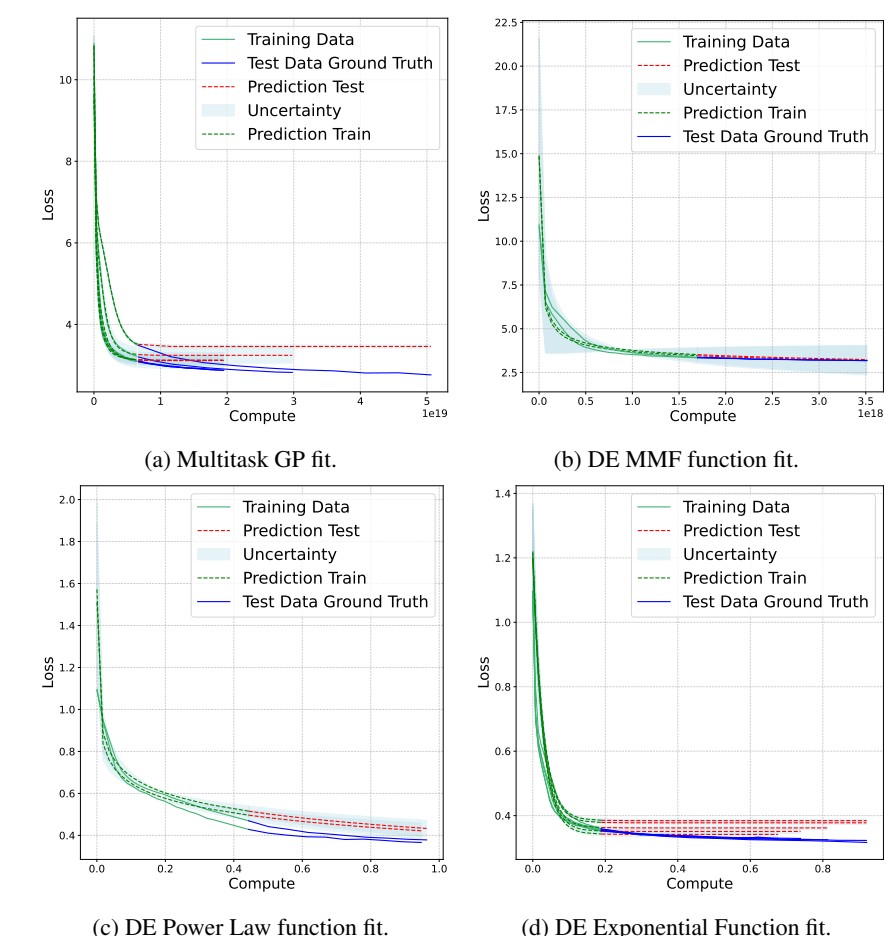

(a) Multitask GP fit.        (b) DE MMF function fit.

(c) DE Power Law function fit.        (d) DE Exponential Function fit.

Figure N.1: While the Deep Ensembles can only approximate the ground truth curve shape, the GP model fits the training data accurately and the choice of the kernel function enables the extrapolation according to the curve shape.

## O  A-PRIORI CREATION OF THE NANOGPT DATASET VS. AN ONLINE SETTING

Any collection of models, such as the nanoGPT variants, is necessarily a discrete, finite set determined by architectural choices (e.g., embedding dimensions, number of layers). In our case, the dataset was constructed by training 90 different models.

There are two main reasons for creating this dataset a priori:
(a) It provides a fixed ground truth for later evaluation, and
(b) It improves computational efficiency.

Importantly, our algorithms do not modify the models to be trained. Instead, they determine which models are selected and how long each is trained, given the allocated compute budgets. In practice, we extract the specific segment of each learning curve that corresponds exactly to the training length implied by the budget at a given point in time.

As a result, the learning curves are identical to those that would be obtained in an online setting interleaved with the algorithms (with the same hyperparameters and convergence behavior). However, the a-priori approach is substantially more resource-efficient, particularly when running multiple algorithms and budget scenarios.

## P    ON THE REGRET ANALYSIS IN SECTION 4.2

Regret quantifies how much worse a given algorithm performs compared to the best possible strategy in hindsight, with regret= $0$ indicating optimal performance. In our setting, the "best possible strategy" corresponds to the lowest loss achievable by any model within the current set. Thus, we compare against an oracle (rather than a baseline) in each run, and report results averaged over $100$ runs.

For context, typical loss values for nanoGPT models range from approximately $10.9$ (at initialization) to $2.6$ (best converged model). Within this range, a regret of $0.41$ provides an interpretable measure of suboptimality. However, no explicit universal upper bound can be stated, as the attainable loss depends on the specific run and model subset considered.

## Q    AREA BETWEEN CURVES

The Area between Curves (AbC) is illustrated in Figure Q.1. In order to apply this measure, one needs to define an AbC range of interest. In this case, it is chosen as $10^{13}$ to $10^{23}$ FLOPs (shaded area).

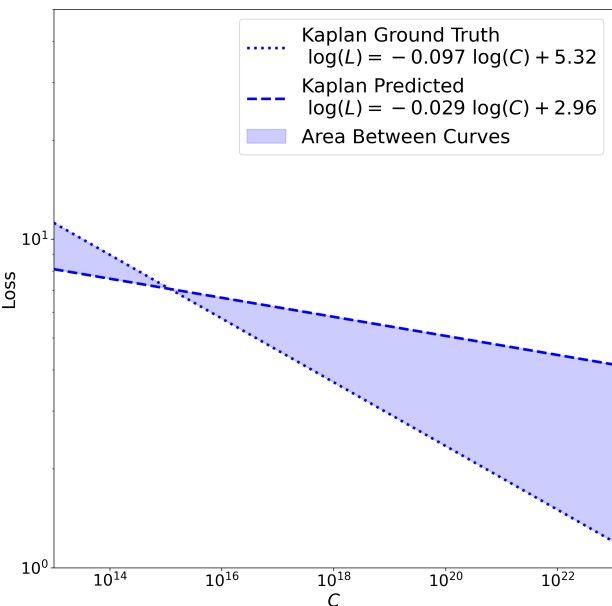

Figure Q.1: Area between Curves (AbC) for the compute range of interest being $10^{13}$ to $10^{23}$ FLOPs (shaded area). The ground truth scaling law according to Kaplan (Pearce & Song, 2024) is compared to an example of a predicted LC.

# R    SCALING LAW ANALYSIS

Figure R.1 illustrates how we obtained the scaling law on the full dataset. Figure R.2 shows a comparison of the scaling law obtained after budget allocation according to SH (a) and SH LMC (b). In both cases we are very close to the ground truth scaling laws. Notice, SH LMC provides a lower test loss value in addition. The full learning curves are shown in light blue. The learning curves obtained after training the models with SH (a) or SH LMC (b) allocation is shown in green. The ground-truth scaling law fitted to the complete learning curves is shown as a dashed blue line, while the ground-truth scaling law fitted to the entire dataset (see Figure R.1) is shown as a solid blue line. The scaling law fitted to the obtained learning curves (solid green) is shown as a dotted blue line. Note, using SH LMC (b), the obtained scaling law (dotted blue) is closer to both ground truth scaling laws compared to using SH (a).

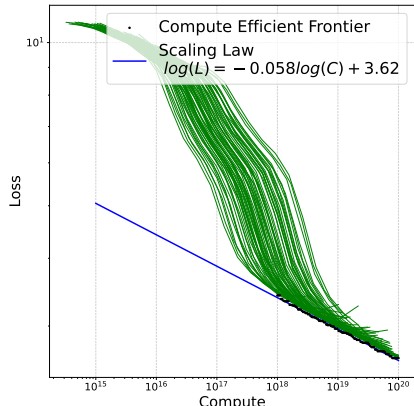

Figure R.1: Scaling Law for the full nanoGPT dataset.

# S    EXTENDED LITERATURE

The problem we consider is orthogonal to the problems of hyperparameter optimization or hyperparameter transfer for ML models considered by e.g. Kadra et al. (2023) and Yang & Shami (2020). We are not searching for the best performing model, but aim to obtain a set of loss-compute LCs suited for SL research. Obtaining the model leading to the lowest loss-compute value among all available models is beneficial as values closer to the loss-compute frontier aid the regression fit to this frontier (see also Figure R.2).

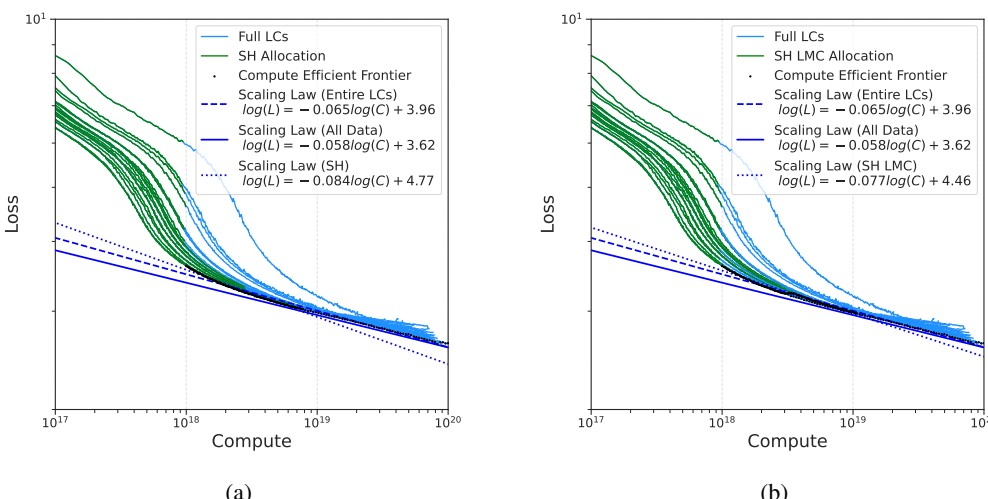

(a) (b)

Figure R.2: Scaling laws assuming the allocated budget is $10^5$ petaFLOPs and the compute region of interest is between $10^{18}$ and $10^{20}$ petaFLOPs.
**a)** The obtained scaling law after allocation according to SH in comparison to the two ground truths scaling laws. The first one is the scaling law assuming fully trained learning curves are available (entire LCs, i.e. green and light blue part is available) and the second one is the scaling law obtained when the whole dataset is available (All Data).
**b)** Scaling law after allocation according to SH LMC in comparison to the two defined ground truth scaling laws.

A conventional Bayesian Optimization approach is not well suited to our setting. This is because our objective extends beyond identifying the parameter configuration that minimizes loss. In particular, we operate over a fixed set of models ($\mathcal{M}_0$), rather than a continuous hyperparameter space. Our method must allocate a fixed compute budget across all models in $\mathcal{M}_0$, with the dual aim of (a) identifying the model that achieves the lowest loss under this budget constraint, and (b) generating a representative collection of learning curves. The latter is essential for fitting the scaling law, which forms a central objective of our study.

As discussed in the introduction and related work, our study builds on prior findings that demonstrate the suitability of surrogate models for learning-curve extrapolation across tasks and model types, primarily in the context of hyperparameter optimization. Section 3 outlines the reasoning behind our kernel choices and related parameters for the LMC model.

To further motivate the applicability of such approaches across a broader range of tasks and models, we note additional relevant works. Learning-curve extrapolation has been studied in the contexts of data acquisition, early stopping, and early discarding (Viering & Loog, 2022; Turan et al., 2025; Mohr et al., 2022). In machine translation, learning-curve prediction was considered by Kolachina et al. (2012). Earlier work has also examined curve prediction for decision trees (Kolachina et al., 2012); CNNs, fully connected networks, linear regression, and variational autoencoders (Klein et al., 2016); and, more recently, Transformers (Kadra et al., 2023).

A variety of functional forms have been explored for extrapolation, including exponential, logarithmic, and power-law functions (Viering & Loog, 2022). Probabilistic approaches related to our DE models have also been proposed, such as ensembles of parametric functions (Domhan et al., 2015) and parametric functions combined with Bayesian neural networks (Klein et al., 2016).

We focus on a low-budget, resource-constrained setting and do not assume access to extensive computational resources such as the 1,000,000 A100-GPU-hours reported by Le Scao et al. (2022). Our objective therefore differs: we study resource allocation strategies designed to make efficient use of limited compute. In contrast, Le Scao et al. (2022) train and evaluate many models at the 1.3B parameter scale before deriving a scaling law and extrapolating to larger models. Rather than adopting this exhaustive approach, we allocate resources strategically, estimate a scaling law, and assess how closely our method approximates the law that would be obtained under an unlimited budget. This is evaluated quantitatively through the AbC values, which measure how well the scaling law is captured in our constrained setting. Finally, as noted in the limitations of Le Scao et al. (2022), concurrent work by Hoffmann et al. (2022) identified more optimal scaling laws. For this reason, we base our analysis on their functional forms instead.

## T ON THE PERFORMANCE OF SH DE VS. SH LMC

SH DE methods performed slightly worse than SH LMC in our experiments (Table 2). The DE approach incorporates knowledge about model size, whereas LMC leverages a correlation term.

Gaussian Processes can be tightly fitted to the observed learning curves, and the combination of inter-curve correlations with a suitable kernel choice enables effective extrapolation. This explains the good performance of the LMC model.

In contrast, DE-based methods present additional challenges. Our DE relies on MLPs, which must first be trained on the available data before extrapolation is possible. In our setup, the number of models per experiment is limited (typically 5–20), which constrains the training data available for the MLP. This introduces two difficulties: (a) the limited data restricts generalization, and (b) the MLP may struggle to learn functional representations that capture the diverse shapes of learning curves.

As shown in Figure C.1, the green learning curves exhibit significant variability: some plateau at high loss values, others at low loss values, and some do not plateau at all. Capturing such variability requires sufficient training data to reliably learn functional coefficients, which is often unavailable in low-resource settings.

Despite these limitations, it is noteworthy that the DE-based surrogate models still achieve performance comparable to LMC (Table 2). This is encouraging, suggesting that in larger-scale experimental setups, DE-based methods could potentially outperform the LMC variant.

## U    MORE INFORMATION ON THE SIMULATION-BASED EXPERIMENTS

We report results from our real-world studies using the nanoGPT dataset, described in Section 4.1. These experiments involve 90 Transformer models of varying sizes, up to 1.5B parameters, all trained on the real-world Fineweb-edu dataset. This provides a sufficiently diverse set of model variants for analysis.

In addition, the synthetic datasets introduced in Sections 4.1 and 4.3 add significant value, especially under limited computational budgets. They not only complement the real-world experiments but also act as effective testbeds for exploring compute-intensive scenarios that would otherwise be infeasible.

For the synthetic dataset used in Section 4.1, we rely on the scaling law parameters determined by Hoffmann et al. to emulate a larger set of experiments in regions beyond our feasible compute range. Importantly, our model does not adopt the same functional form or parameters used to generate those curves. Thus, no prior information is transferred that could compromise the validity of the experimental setup.

Finally, to more closely reflect real-world conditions, we also study the effects of different noise types (Figures 3 and 4).

## V    USING OUR INSIGHTS IN PRACTICE

If simplicity and lowest-possible algorithmic cost and speed are paramount: SH is preferred. This method is simpler, even 'cheaper' than SH LMC.

If the model with lowest loss shall also be obtained for further use (e.g. for the finetuning on downstream tasks, etc): SH LMC is preferred, because this method yields a lower-loss model as demonstrated by reduced loss-regret.

If the allocated budget cannot efficiently cover the loss-compute frontier in the compute region of interest for the scaling law (e.g. too little budget available to further train larger models): An LMC prediction step in addition to SH LMC can effectively extrapolate the LCs to obtain more predicted points on the frontier, and in this way further improve the scaling law (Section 4.3, Figure 5 and Table 4).

## W    ON LIMITED COMPUTE BUDGET

Our limited computational budget naturally imposes constraints on the scope of our research. Nevertheless, we believe that our work provides comprehensive and valuable insights by exploring a range of feasible settings. These include experiments with multiple surrogate models, studies on real-world datasets where possible, and synthetic data based on prior compute-intensive work, which allows us to emulate otherwise infeasible scenarios.

Especially in light of the computational budget limitations we face, which are common across the research community, we hope our work highlights the importance of computational efficiency and inspires greater attention to its role in scaling law research.

## X    ON ADDITIONAL EXPERIMENTS

Under less resource-constrained conditions, extending our experiments would further strengthen this work. We view such directions as promising opportunities for future research.

Nevertheless, we believe our study already provides comprehensive and valuable insights by conducting experiments across a diverse range of datasets and perspectives that are feasible within a compute-limited setup. These include evaluations with multiple surrogate models, analyses on real-world data where possible, and synthetic data derived from prior compute-intensive studies to emulate scenarios that would otherwise be infeasible. To the best of our knowledge, this is the first work to address scaling law research through strategic budget allocation aimed at maximizing efficiency.

Given that computational budget limitations are shared by many researchers, we hope our work highlights the importance of computational efficiency and encourages the community to place greater emphasis on its role in scaling law research.

# Y    ON THE USE OF LLMs

The authors used ChatGPT to assist with language refinement, in a manner similar to consulting an online dictionary. We confirm that all content and ideas are entirely our own and were not generated by any language model.

# Z    LIMITATIONS, COMPUTE RESOURCES AND IMPACT STATEMENT

**Limitations.**    Increased computational resources and access to larger datasets would provide deeper insights into this research direction. However, due to computational constraints, our experiments were conducted on a limited set of learning curves.

The current compute range of interest is between $10^{18}$ and $10^{20}$. Given more computational resources a learning curve set with a wider spread of learning curves would be interesting to investigate.

**Compute Resources.**    To obtain the learning curves of the nanoGPT models we used four NVIDIA A100 (80GB) GPUs. Successive Halving experiments were conducted on CPUs (Intel(R) Xeon(R) Gold 6448H).

**Impact Statement.**    The experiments in our paper assume a limited budget setting. Note that while the limit in our experiments could be considered on the lower end given our computational constraints, any real-world setups will still have a limited budget; even if this limit might be quite high. Hence, assuming a fixed budget seems natural – and under this consideration, only a fixed number of models could be sufficiently trained, as the budget has to be allocated in some way. Therefore, as shown in Table 3, compared to the costs that occur to obtain the full nanoGPT dataset, or train the LCs of interest fully, SH or SH LMC can provide a cost-efficient method for compute resource allocation leading to a obtained scaling law close to ground truth scaling laws.

This means that, even if a large budget is available and many models can be sufficiently trained, our method can allocate this budget in a potentially even better way, freeing up resources to train more models whose learning curves will be closer to the loss-compute frontier – which is essential for scaling laws. In addition, as we demonstrated in Table 4, our method in combination with another LMC for extrapolation can further predict the continuation for even larger computes – providing additional information to determine the scaling law (compare Figure 5 left vs. right).

Despite the scaling law being based on a prediction (which could affect the coefficients), a prediction using UCB and LCB values in addition, will give a range in which the original scaling law can be expected. This adds a quantified measure of uncertainty and will therefore provide further guidance for decision making.

