# OpenReview forum: "Charting the Frontier: How Optimizing Performance Yields Accurate Scaling Laws on a Shoestring"
_ICLR.cc/2026/Conference — Submitted to ICLR 2026_

### Official Review · Reviewer_jZTL · 2025-10-25

**Soundness:** 2
**Presentation:** 2
**Contribution:** 2
**Rating:** 2
**Confidence:** 4

**Summary:**

This paper presents an algorithm inspired by the bandit literature for optimizing compute allocation to fit a scaling law. Experiments show the possibility of extrapolation of scaling laws using less compute than a “uniform allocation” baseline.

**Strengths:**

1. It is a nice idea to leverage ideas from the bandit literature to optimize compute allocation while fitting scaling laws.
2. The paper provides good evidence that their particular surrogate decisions work well within the framework of evaluation that they set up.
3. The added benefit of uncertainty estimates is a nice side benefit.

**Weaknesses:**

1. The clarity of the paper not good. From figure 1, the curves are not properly labeled. Most of the algorithm details are not clear from the main text, the reader really needs to do a lot of work to figure out what is going on. Line 5 of the main algo is not even properly defined. In general, setting up the bandit problem that motivates the paper is really hard to grok from a quick read.
2. The optimization problem in equation 2 does not make sense as a proxy since the target problem in Eq 1 is maximizing for a full set of compute budgets, but the proxy only selects a single model. This then becomes a top-k in the algorithm, but the motivation here is very unclear.
3. The whole premise of the goal of the scaling law seems to be wrong. In general, the point of fitting a scaling law is to predict what happens when we do a final run with a much larger compute budget. We don’t care about the “goodness of fit” of the curve on lesser compute points per se and we also don’t care about trying to find the minimal loss from a set of models (it will be the largest one). The goals are twofold: (1) predict what will happen when we do a run with C_large compute so we know the ROI of a larger run, and (2) compare two different methods A and B at small scale but extrapolate which one would be better at C_large in the final run. This paper does not seem to attack either goal.
4. The scaling law that is fit throughout the paper has no irreducible error term. This implicit assumption that infinite compute would give 0 loss seems that it would cause issues scaling the method up as is.
5. The main “uniform allocation” baseline is a bit unclear and seems that it could be a sort of straw man. Papers like Hoffman et al. are not trying to do efficient prediction of the target loss, they are trying to present the scientific finding of the robust scaling law, so using their methodology exactly as a baseline for efficiency seems to miss the point a bit.

**Questions:**

1. What is hat L in line 5 of algorithm 1?

---

> ### Author Response · Authors · 2025-11-23
> **Response to Reviewer jZTL (part 1/2)**
>
> Thank you for your helpful review. Before we address your questions individually in the following, we would like to clarify a potential misconception:
>
> **Clarification of potential misunderstanding about the core contributions and goals of our work**:
> In prior scaling law work, the emphasis is on scientific characterization of the scaling relationship (e.g. Kaplan et al., 2020; Hoffmann et al., 2022). By contrast, our work focuses on *efficient generation of the points* used to fit such scaling laws.
>
> Specifically, prior works tend to fully train all candidate models to convergence.
> $\rightarrow$ Our approach instead predicts early which models are unlikely to improve, and iteratively prunes the candidate set until only the most promising model remains.
> $\rightarrow$ This allows us to *allocate a fixed compute budget more efficiently* while producing approximately the same scaling points that would be used in a classical analysis.
>
> $\Rightarrow$ Importantly, we do not modify the scaling law functional form; rather, we aim to provide a more compute-efficient methodology to obtain the same data for fitting.
>
> ---
> **[W1] Clarity**
> > *From figure 1, the curves are not properly labeled.*
>
> Thank you for pointing this out. In Figure 1, the variable $C$ denotes Compute, as described in the text. For clarity, we have now labelled it explicitly as 'Compute $C$'. The y-axis label 'Loss' is displayed on the left and applies to all panels in the row, following standard convention.
> > *algorithm details are not clear from the main text, the reader really needs to do a lot of work to figure out what is going on*
>
> Lines 197-224 provide a high-level overview of the method, and Algorithm 1 is intended to present the core procedure in a concise but complete form, given the space constraints of the initial submission.
> $\rightarrow$ As noted in line 219, we include a detailed line-by-line walk-through of both Algorithm 1 (core SH procedure) and Algorithm 2 ('obtain\_LCs'; surrogate-based prediction) in Appendix B.1 and B.2, where all individual steps are explained in detail.
>
> $\Rightarrow$ If you find it helpful, we are happy to move some of these details into the main text in the revision to improve clarity.
>
> ---
> **[W2] Proxy task**
> > *equation 2 does not make sense as a proxy [...]*
> > *proxy only selects a single model. This then becomes a top-k [...]*
>
> We believe there may be a misunderstanding about the role of the proxy objective in Eq. (2):
> $\rightarrow$ The purpose of Eq. (2) is *not* to select a single final model; rather, it provides a tractable surrogate/proxy for the intractable objective in Eq. (1), which would require fully training all models to convergence.
>
> As discussed in lines 182-189, the true objective (Eq. 1) cannot be solved under realistic compute budgets, and even defining $G(M)$ would require a gold-standard scaling law obtained only by fully training every model.
> $\rightarrow$ Eq. (2) therefore acts as a proxy that allows us to allocate budget across models in a principled way.
>
> Importantly, the algorithm does *not* use Eq. (2) to choose only a single model:
>
> $\Rightarrow$ During Successive Halving, we iteratively prune a set of models using top-k, and continue training the surviving subset each round. This process produces *a set* of partially and fully observed learning curves (not just one curve), which we then use to fit the final scaling law.
>
> In other words, optimizing the proxy objective naturally produces the *set of curves* required to approximate Eq. (1) (which is then used for the scaling law fit).
>
> ---
> *... continued in part 2...*

---

> > ### Author Response · Authors · 2025-11-23
> > **Response to Reviewer jZTL (part 2/2)**
> >
> > **[W3] Goals/Scaling law**
> > > *The whole premise of the goal of the scaling law seems to be wrong. [...] the point of fitting a scaling law is to predict what happens when we do a final run with a much larger compute budget. [...]. The goals are twofold: (1) predict what will happen when we do a run with $C_{large}$ compute so we know the ROI of a larger run, and (2) compare two different methods A and B at small scale but extrapolate which one would be better at $C_{large}$ in the final run. This paper does not seem to attack either goal.*
> >
> > We appreciate your summary of classical scaling law objectives. As noted in our 'clarification' paragraph above, our work is *complementary* to these goals: we do *not* aim to propose a new scaling law or change its functional form.
> >
> > $\Rightarrow$ Instead, we focus on *efficiently generating the points* used to fit a scaling law.
> >
> > While classical scaling studies emphasize extrapolating performance at very large compute budgets, practical use often requires making *informed decisions under limited budgets*. By predicting early which models are unlikely to improve, our approach allows us to allocate compute more effectively, producing approximately the same scaling points with substantially lower resource consumption.
> >
> > $\Rightarrow$ In this sense, we are providing a compute-efficient methodology that supports the same goals of ROI estimation and method comparison, but in a more resource-conscious manner.
> >
> > ---
> > **[W4] Irreducible error**
> > > *no irreducible error term*
> >
> > Thank you for pointing this out.
> > Our analyses in this work are mainly centered around finite and practically-constrained compute budgets. Thus, the absence of an explicit irreducible error term does not affect the insights of our method.
> >
> > $\Rightarrow$ Importantly, our pruning-based approach is *agnostic* to the scaling law form and can be equally applied to fits that include an irreducible term if desired.
> >
> > For reference, Appendix H (Eq. (4) and Table H.1) includes the irreducible error term $E$ when generating our synthetic experiments (Hoffmann et al. (2022); Besiroglu et al. (2024)). This term represents the minimum achievable loss and ensures that losses do not drop below the irreducible floor.
> > $\rightarrow$ While the main text uses a linear log-log scaling law for simplicity, a power-law plus constant naturally asymptotes to $E$ at very large compute, producing the expected plateau.
> >
> > We also note that our work does not target a specific scaling law per se: we follow the common practice of fitting a functional form to the loss-compute frontier (Pearce and Song, 2024; Kaplan et al., 2020), while our contribution focuses on efficiently generating the points needed to fit such laws.
> >
> > $\Rightarrow$ Including an irreducible error term is orthogonal to this contribution and can be incorporated without modifying the method.
> >
> > We will add a note in the main text, and have further expanded Appendix H to highlight this point.
> >
> > ---
> > **[W5] Uniform baseline**
> > > *'uniform allocation' baseline is a bit unclear[...]*
> >
> > We would like to clarify that the purpose of our uniform allocation baseline is not to replicate the scientific objectives of prior work, but to provide a meaningful reference for compute allocation efficiency. Our approach, by predicting early which models are unlikely to improve, iteratively prunes the candidate set and concentrates resources on the most promising model(s).
> >
> > $\rightarrow$ In this sense, the uniform allocation baseline is a fair comparison point: it represents the naive strategy of allocating roughly equal compute to all candidates, allowing us to quantify the practical gains of our efficiency-oriented methodology.
> > $\rightarrow$ Using prior scaling law methods as a baseline does not aim to reproduce or challenge their scientific findings, but to benchmark our strategy in a way that highlights compute efficiency while still aiming to produce the same scaling points for analysis.
> >
> > ---
> > **[Q1] Algorithm line 5**
> > > *What is hat L*
> >
> > We thank you for pointing this out. The variable $\hat{\mathcal{L}}$ in line 5 of Algorithm 1 collects all predicted continuations of learning curves when using SH with surrogates.
> >
> > In other words:
> > $\mathcal{L}$ collects all 'trained' curves, *excluding* the continuations predicted by surrogate models ($\hat{\mathcal{L}}$).
> > $\rightarrow$ Hence, we update by $\tilde{\mathcal{L}} \setminus \hat{\mathcal{L}}$ (since $\tilde{\mathcal{L}}$ contain all curves, trained and predicted).
> > $\rightarrow$ Details can be found in Appendix B.2 'obtain\_LCs' (Algorithm 2), particularly descriptions of lines 17 and 18.
> >
> > $\Rightarrow$ We will update the manuscript to include the proper definition of the $\hat{\mathcal{L}}$ in the main text.
> >
> > ---
> > ---
> > We hope our answers and additional explanations have clarified your questions.
> > If you have any further queries or any points remain unclear, please let us know and we are happy to answer them!

---

> > > ### Comment · Area_Chair_AGqn · 2025-11-25
> > > **Re goal**
> > >
> > > If you focus on efficiently training scaling laws, wouldn't citing existing papers on scaling laws that try to recycle points or suggest more efficient uses of points in the scaling laws contextualize the work better? It feels like the citations lack a lot of that kind of works (hagele's work .
> > > e.g.
> > > https://arxiv.org/abs/2412.06540 https://arxiv.org/abs/2405.10938 (good paper, but actually only a horizontally useful) https://arxiv.org/abs/2410.11840 probably many others. Maybe the only one is https://arxiv.org/abs/2405.18392 (which is a good and worth citing)
> > > Seems like such comparisons are quite lacking and might ease comparisons, (although the motivation does support this point of view anyway).

---

> > > > ### Author Response · Authors · 2025-11-26
> > > > **Thanks! -> Revised related work**
> > > >
> > > > Thank you for pointing out this angle and these works! We agree that while our motivation outlines our goal for efficiency, making this *more explicit* and incorporating more related works will definitely strengthen this point; and hopefully make it easier for the reader to grasp and better place our work in the context of others.
> > > >
> > > > We currently contrast to Hagele et al. in the second-last sentence in our related works (paragraph 'Scaling Laws', ll.128-131):
> > > > > *Hagele et al. (2024) proposed an approach to reduce the computation requirements to obtain SLs by using alternative model training techniques: constant learning rates with cooldowns and stochastic weight averaging. Our work is orthogonal to this approach, and is focused on optimizing the computational budget allocation for training.*
> > > >
> > > > $\Rightarrow$ To further improve clarity, we have now extended this and incorporated an additional 'explicit' paragraph on **Efficiency in Scaling Law Research**, outlining related approaches as:
> > > >
> > > > > **Efficiency in Scaling Law Research.**
> > > > >*Hagele et al. (2024) proposed an approach to reduce the computation requirements to obtain SLs by using alternative model training techniques: constant learning rates with cooldowns and stochastic weight averaging. Our work is orthogonal to this approach, and is focused on optimizing the computational budget allocation for training.*
> > > > >*Several other recent works have focused on efficiently deriving scaling insights across and/or within different model families, mainly based on results reported for various benchmarks.*
> > > > >*Ruan et al. (2024) pioneer this direction of 'observational' or 'benchmark scaling laws', utilizing dimensionality reduction via PCA to demonstrate that model performance can be modeled in a low-dimensional space of abstract 'principal capabilities'. Their method is primarily designed to interpret the skills of already trained LLMs and predict complex downstream performance based on simpler proxy metrics. Building on this latent-variable perspective, Polo et al. (2025) propose a conceptually similar but more parametric framework in which benchmark performance is modeled by a small set of explicitly interpretable 'skills' (e.g. reasoning). The authors also specifically design their approach to predict the performance of 'future', yet untrained models across benchmarks and model families.
> > > > A somewhat orthogonal line of work by Choshen et al. (2025) provides a comprehensive guide to improving the robustness of traditional loss scaling law estimation by analyzing and mitigating the effects of variability across training regimes and model families within existing learning curves. However, these methods focus on extracting insights about scaling behavior from existing evaluation or training data; whereas our work in contrast focuses on optimizing the allocation of compute to actively generate the underlying learning curves themselves.*
> > > >
> > > > $\Rightarrow$ We have also incorporated this new and extended related works section into our manuscript (visualized in 'blue' textcolor for clarity).

---

### Official Review · Reviewer_qfWJ · 2025-11-01

**Soundness:** 3
**Presentation:** 2
**Contribution:** 2
**Rating:** 6
**Confidence:** 3

**Summary:**

This paper investigates using successive halving (SH) combined with surrogate models to efficiently estimate the compute-optimal scaling laws in training language models. On both synthetic and real experiments, the proposed method outperforms naive uniform allocation and SH alone.

**Strengths:**

- There is very little published work on efficiently and accurately fitting the scaling law, despite its various challenges such as needing to train models to the compute-optimal frontier while minimizing the overall experiment cost. This work thus touches on an important, open problem.
- The combination of synthetic and real experiments provides good emprical validation to the proposed method.

**Weaknesses:**

- The noise models do not appear well-motivated by theoretical or empirical study on learning curves [1,2,3], which can provide more sensible models for the correlation than Brownian motion, for example.
- There appear to be no details on how hyperparameters such as learning rate, initialization, and weight decay is optimized. These hyperparameters are crucial for producing reliable scaling laws, and there are principled approaches such as [4] on how their optimal values scale with compute that do not seem to have been adopted in this work.
- The Chinchilla scaling law [5] shows that to reach the compute-optimal frontier, 20 tokens should be trained on per parameter. This means the 1.5B model used in this paper shoud be trained for approximately 3e20 FLOPs, 3x the maximum compute shown in Figure 2. Yet the authors are able to identify the compute-optimal frontier at this scale, which suggests something is off with the experiment.
- Based on the shape of the learning curves, I'm guessing no learning schedule or weight decay is used. Recent work [6] has shown that when using a learning schedule or weight decay, performance rankings can vary dramatically from early to late times and can be hard to predict due to non-intuitive learning curve shapes, potentially significantly reducing the effectiveness of the proposed approach.

[1] Bordelon et al., A Dynamical Model of Neural Scaling Laws

[2] Paquette et al., 4+3 Phases of Compute-Optimal Neural Scaling Laws

[3] Qiu et al., Scaling Collapse Reveals Universal Dynamics in Compute-Optimally Trained Neural Networks

[4] Yang et al., Tensor Programs V: Tuning Large Neural Networks via Zero-Shot Hyperparameter Transfer

[5] Hoffmann et al., Training Compute-Optimal Large Language Models

**Questions:**

- Does the proposed method work when following best training practice, such as using a cosine or WSD learning rate and weight decay?

---

> ### Author Response · Authors · 2025-11-23
> **Response to Reviewer qfWJ (part 1/2)**
>
> We thank you for your constructive review and address your questions in the following:
>
> **[W1] Noise models**
> > *noise models do not appear well-motivated by theoretical or empirical study on learning curves [1,2,3]*
>
> [1,2,3] highlight a range of noise sources and training-phase behaviors that can arise in large-scale training; While they provide valuable qualitative insights into how noise may manifest *in general*, the concrete characteristics of that noise (variance, temporal correlation, drift, ...) can vary substantially depending on architecture, data distribution, optimization details, and training schedules.
>
> $\rightarrow$ Our goal was not to replicate the exact noise patterns of any specific setup, but to instead cover the broad space of commonly-observed behaviors/patterns.
> $\rightarrow$ We selected a set of well-established stochastic processes whose behaviors *collectively cover* a diverse set of qualitative noise patterns (many of which also described in your references):
> $\rightarrow$ In particular, we used (i) additive white Gaussian noise as a baseline for uncorrelated perturbations, (ii) Brownian-type processes to capture cumulative and drifting effects, and (iii) Ornstein-Uhlenbeck noise to represent correlated, mean-reverting fluctuations.
>
> $\Rightarrow$ Importantly, these models are straightforward to sample from, computationally efficient, and allow us to systematically probe robustness across a broad range of noise scales.
>
> ---
> **[W2] Hyperparameters**
> > *no details on how hyperparameters*
>
> We have now added a detailed overview to Appendix M, Table M.1; (also see our answer to Reviewer *t952* [Q2] for more info)
> > *principled approaches such as [4] [...] do not seem to have been adopted in this work*
>
> We agree that principled hyperparameter transfer methods such as $\mu$Transfer [4] can improve optimization efficiency as models scale.
> $\rightarrow$ We acknowledge that our choice of using a *fixed* set of hyperparameters across model sizes (similar to Kaplan et al.) could introduce scale-dependent artifacts (e.g., slight vertical or horizontal shifts, mild instabilities), whereas $\mu$Transfer is expected to reduce these effects and make learning curves align more closely across scales.
>
> $\Rightarrow$  However, this would *only strengthen* the assumptions underlying our method, as it is expected to increase cross-scale similarity of learning curves rather than undermining it.
>
> While we chose a single consistent hyperparameter recipe for all scales to keep the comparison controlled in this work, we agree that integrating $\mu$Transfer is a valuable direction for follow-up work (and will add corresponding pointer to the paper).
>
> *Additional note:*
> Recent work by Bjorck et al. (ICLR 2025) demonstrates that even when using $\mu$P, the 'optimal' hyperparameter choice (particularly learning rate) seems to also depend on other factors like token training horizon;
> $\rightarrow$ Given that our method is based on extrapolating learning curves via surrogates (which can be adapted to various curve shapes and offsets), we expect that these recent as well as future insights can be directly incorporated into our method, as they are mostly orthogonal to our focus of improving efficiency via pruning the set of models during training.
>
> ---
> **[W3] Chinchilla assumptions vs. ours**
> > *compute-optimal frontier, 20 tokens [...] per parameter. [...] the 1.5B model should be trained for approximately 3x the compute*
>
> We thank you for raising this point. We agree that, *in principle*, the Chinchilla scaling law implies that a 1.5B-parameter model would require roughly 3x more compute (and correspondingly more data) to reach the global compute-optimal point under *their* assumptions.
> $\rightarrow$ Our setup is constrained by a significantly smaller data budget (max 10B tokens total) and substantially lower compute. Consequently, the 20 tokens-per-parameter guideline is not attainable in our setting.
>
> $\Rightarrow$ The compute-optimal frontier we report is therefore the *optimal point within our fixed, limited data-compute regime*, not the Chinchilla-optimal point that would require access to substantially more training tokens (and compute).
>
> $\Rightarrow$ Note: Identifying this constrained-optimal frontier is *still valid*, internally consistent and very useful in practice: it reflects the best achievable trade-off given the data and compute *actually available*, rather than the theoretical optimum in the unlimited-data regime.
>
> We will add a clarifying sentence to our manuscript to explicitly state this constraint:
> *For clarity, we note that all reported compute-optimal points across our real-world nanoGPT experiments are identified within our fixed 10B-token training budget; they do not represent the global Chinchilla-optimal points, which would require significantly larger data and compute, and are outside the scope of our experiments.*
>
> ---
> *...continued in part 2...*

---

> > ### Author Response · Authors · 2025-11-23
> > **Response to Reviewer qfWJ (part 2/2)**
> >
> > **[W4] & [Q1] LR scheduler**
> > > *Based on the shape of the learning curves, I'm guessing no learning schedule or weight decay is used.*
> > > *[...] performance rankings can vary [...] from early to late times*
> >
> > > *Does the proposed method work when following best training practice, such as using a cosine or WSD learning rate and weight decay?*
> >
> > The apparent 'absence' of LR scheduling in the figures is likely because we plot them on a log-compute scale, which visually compresses the long-tail portion of the cosine-decay schedule.
> > $\rightarrow$ All of our experiments use a *cosine-decay* learning-rate scheduler with linear warmup and decay to 10\% of the peak LR. We do not use weight decay since our early experiments showed that additional regularization provided little benefit for our (comparably small) models.
> >
> > $\rightarrow$ Importantly, our LR schedule is parametrized with respect to the maximum training duration a model could experience if it passes all SH rounds.
> > $\Rightarrow$ This ensures that each model’s learning curve evolves exactly as it would under a standard fully trained run; SH only determines how long the model is allowed to continue, not how its LR dynamics evolve up to that point. Our method therefore remains compatible with such dynamics.
> >
> > $\Rightarrow$ This setup mitigates the LR-related sensitivity and late-stage ranking changes highlighted in prior work, as SH only affects termination/continuation of training; not the training dynamics themselves.
> >
> > ---
> > ---
> > We hope our answers addressed all your questions.
> > If any points remain unclear, please let us know and we are happy to clarify them.

---

### Official Review · Reviewer_2jwe · 2025-11-01

**Soundness:** 3
**Presentation:** 3
**Contribution:** 3
**Rating:** 6
**Confidence:** 3

**Summary:**

This paper presents a cost-efficient framework for estimating neural scaling laws under computational constraints. The authors adapt the Successive Halving (SH) algorithm to allocate compute across model families and further enhance it with surrogate models that predict future learning-curve trajectories. Empirical evaluations on both synthetic and real-world nanoGPT datasets demonstrate up to 98.7% compute savings while maintaining accurate scaling-law estimation.

**Strengths:**

1. This paper tackles an important problem: efficient estimation of scaling laws under limited compute budgets, directly relevant to LLM development and deployment.

2. This paper has demonstrated remarkable computational efficiency, reducing compute usage by up to 98.7%.

3. This paper shows consistent performance across synthetic and real-world datasets.

**Weaknesses:**

1. The method’s performance strongly depends on the surrogate model’s fidelity in capturing learning-curve dynamics; Deep Ensembles in particular show instability on real data.

2. The framework optimizes only validation loss, without considering multi-objective trade-offs such as downstream generalization.

3. Experiments are limited to nanoGPT, leaving scalability to larger LLMs unverified.

**Questions:**

1. Could the proposed framework be extended to incorporate additional objectives (e.g., generalization or transfer performance) to better align with real-world model training considerations?

---

> ### Author Response · Authors · 2025-11-23
> **Response to Reviewer 2jwe**
>
> Thank you for your constructive review.  We address your points individually in the following:
>
> **[W1] Dependence on surrogate**
> > *performance strongly depends on the surrogate model’s fidelity*
>
> We agree that surrogate fidelity is important for capturing learning-curve dynamics.
> $\rightarrow$ To address this, we adopt a modular approach in our work and evaluate multiple surrogate types, including non-parametric Gaussian Processes (SH-LMC) and parametric Deep Ensembles.
> $\Rightarrow$ This modular design allows practitioners to easily swap in alternative surrogates or incorporate a-priori knowledge from previous experiments (or experts), offering flexibility to adapt to specific datasets, tasks or model families.
>
> ---
> **[W2] \& [Q1] Extension to additional objectives**
> > *optimizes only validation loss, without considering multi-objective trade-offs [...]*
> > *Could the proposed framework be extended to incorporate additional objectives [...]*
>
> This is indeed quite an interesting suggestion.
>
> $\rightarrow$ Yes; while our current framework uses validation loss (expected perplexity) as the main criterion for SH pruning, it is modular and could *in principle* incorporate other objectives, such as downstream task performance or transfer metrics.
> $\rightarrow$ Recent work (Chen et al., TMLR 2025) builds on observations in previous work that once pre-training loss reaches a critical threshold, there is a strong correlation between pre-training loss and downstream performance; The authors propose a two-stage approach, mapping 1) FLOPs $\rightarrow$ Loss and then 2) Loss $\rightarrow$ Performance;
>
> $\Rightarrow$ We expect our surrogate predictions could be extended to target downstream metrics in a very similar way.
> We note, however, that we haven't run any experiments in this direction, and can only provide 'intuition' here. But we consider multi-objective or downstream-aware variants as quite an interesting direction for future work.
>
> ---
> **[W3] Larger models**
> > *Experiments are limited to nanoGPT, leaving scalability to larger LLMs unverified.*
>
> Our limited available computational budget naturally imposes hard constraints on the scope of our research (as we discuss in Appendix W).
> $\rightarrow$ Nevertheless, we believe that our work still provides comprehensive and valuable insights by conducting experiments across various datasets and perspectives that are feasible within a compute-limited setup; including experiments involving multiple surrogate models, studies on real-world datasets where possible, and synthetic data based on prior (compute-intensive) work, which allows us to emulate otherwise infeasible scenarios.
>
> $\Rightarrow$ Especially because of our computational budget limitations (that are shared by many other researchers), we hope that our work can inspire the community to focus more on how computational efficiency can play a much bigger role in scaling law research.
>
> ---
> ---
> We hope our answers addressed all your questions.
> If you have any further queries, please do not hesitate to reach out.

---

### Official Review · Reviewer_t952 · 2025-11-01

**Soundness:** 3
**Presentation:** 3
**Contribution:** 2
**Rating:** 4
**Confidence:** 3

**Summary:**

This paper proposes a resource-efficient approach to estimating compute-loss scaling laws by applying Successive Halving (SH) with and without surrogate models (Gaussian Processes and Deep Ensembles). The key premise is that traditional methods for deriving scaling laws (e.g., Kaplan et al., 2020) are computationally expensive, requiring full training of many models. The authors reformulate the problem as one of optimal compute allocation across a set of candidate models, then use SH to progressively prune models during training. They further enhance SH by predicting future learning curves using surrogate models conditioned on partial data, allowing for more informed resource allocation.

**Strengths:**

Originality: Reframes scaling law estimation as a resource allocation problem and bridges hyperparameter optimization with scaling law research. Using SH and learning curve surrogates in this context is novel to my knowledge.

Clarity and quality: Paper is well-structured with visualizations and thorough appendices. The evaluation is done in multiple different settings, including synthetic experiments and nanoGPT data. Surrogates are well-motivated and tested across tasks.

Significance: Large computational savings for establishing scaling laws.

**Weaknesses:**

The two main discussion points to me seem 1) complexity of the approach and 2) unclear effect on downstream accuracy of the scaling law.

For 1), the training of multiple models across scales is already a complex endeavor, requiring strategies of how to distribute many parallel training runs across a cluster, budget allocation, checkpointing, learning rate schedules, etc. I am skeptical of the overall introduced complexity of the SH + surrogate approach as a blocker for adoption; e.g., continuously stopping and restarting real model runs in an online fashion.

For 2), to me, Table 3 doesn't clearly establish whether the learned scaling laws (under budgeted evaluation) are actually useful for predicting downstream model performance beyond the observed region. In a way, the predictive quality of a scaling law is what matters most for practitioners -- I would rather spend more on running small model ablations to arrive at a precise fit for my future large runs, than save compute with SH to get a suboptimal fit. I will rephrase in the question section below.

**Questions:**

The AbC metric in Table 3 only measures fit error against fully observed curves. How well do the scaling laws—fit using SH+surrogate under a limited budget—predict performance for unseen (larger) models or higher compute budgets? In other words, have you evaluated the extrapolation accuracy of the learned scaling law, or compared it to a baseline scaling law fit on the fully trained models?

Misc.: You simply say you use default hyperparameters for nanoGPT. Could you clarify the exact details, including batch size, learning rate schedule, total tokens, ...etc? Especially the LR schedule will have a strong influence on scaling laws (cf the original Chinchilla observation of the LR decay effect compared to Kaplan), even more so when you perform the model selection in the online fashion to stop and continue training.

---

> ### Author Response · Authors · 2025-11-23
> **Response to Reviewer t952**
>
> We thank you for your helpful review, and address your questions individually in the following:
>
> **[W1] Complexity**
> > *skeptical of the overall introduced complexity [...] e.g., continuously stopping and restarting [...]*
>
> We appreciate your concern about potential complexity. In practice, however, both components of our method, i.e. Successive Halving (SH) and the surrogate fits, introduce negligible overhead compared to the cost of training the models themselves, and do not require heavy stop-restart management:
>
> -  *Surrogate fitting is negligible/amortized*: Our LMC surrogate models take only a few minutes (max) to fit. This cost is several orders of magnitude smaller than training the neural models.
> -  *SH integrates cleanly into existing training setups*:
>   Our method does not require repeatedly stopping and restarting models. It is implemented as a lightweight decision step when the allocated budget for the current round is reached (similar to standard checkpointing).
>   $\rightarrow$ Models that remain in the candidate pool simply continue training normally.
>   $\rightarrow$ Models that fail the SH criterion exit cleanly and free up resources for other models/tasks.
>   $\Rightarrow$ This setup aligns with many existing distributed training pipelines which already run periodic evaluation, checkpointing, etc.
> - *Straightforward parallelization*:
>   $\rightarrow$ Synchronous, within-stage parallelization: All candidates in round $r$ train in parallel for the round’s allocated budget; Synchronization is required only at end of the SH-round, not during training.
>   $\rightarrow$ Optional extension via Asynchronous SH: If any delay/suboptimal resource use is a concern, there exist fully asynchronous SH variants like ASHA (Li et al., 2018) that eliminate global synchronization barriers.
>
> $\Rightarrow$ Additional engineering and compute overhead is therefore low relative to the training itself, and our method fits naturally into most training pipelines.
>
> ---
> **[W2] & [Q1] Extrapolating model performance**
> > *would rather spend more on running small model ablations to arrive at a precise fit[...]*
>
> Prior work has shown that scaling laws estimated solely from a tightly-concentrated set of (e.g. small) models *may not extrapolate reliably* to larger-scale behavior: Hoffmann et al. (2022, Appendix E, Fig. A5) show that fits based on the first, middle, or last third of model sizes yield different projections.
> $\rightarrow$ This suggests that incorporating a broader range of model sizes can improve the stability of extrapolation.
> $\Rightarrow$  Our method is designed precisely to make such broader exploration feasible under a fixed budget. By allowing some mid- and larger-scale models to contribute without fully training all candidates, SH+surrogate helps to obtain scaling-law fits that can better reflect the true frontier.
> > *have you evaluated the extrapolation accuracy of the learned scaling law, or compared it to a baseline scaling law fit on the fully trained models*
>
> We explicitly assess this in the 'Entire LCs SL' column of Table 3:
> $\rightarrow$ While the '*Full Data SL*' AbC metric in Table 3 measures fit error against fully observed curves for the models included in the SH subset, the '*Entire LCs SL*' column directly evaluates extrapolation beyond the observed region.
> $\rightarrow$ Specifically, it compares the scaling law obtained under budgeted SH+surrogate to the one fit on all fully trained models, including the largest models not seen during SH (especially for small budgets $B$ and small model sets $M_0$).
> $\Rightarrow$ This tests whether SH+surrogate preserves the shape of the true scaling relationship across the full model range, and the results indicate that extrapolation accuracy remains strong under our budget constraints.
>
> ---
> **[Q2] Hyperparameters**
> > *clarify the exact details [...] LR schedule will have a strong influence [...]*
>
> Thank you for pointing this out. We have now included all missing hyperparameters in Appendix M, Table M.1.
>
> In brief, we mostly follow the nanoGPT defaults (Karpathy, 2022), including AdamW with default settings; no wd/dropout.
> Lr scheduler: Cosine decay down to 10\%, linear warmup (peak lr $6\times 10^4$)
> *Importantly*:
> $\rightarrow$ Our cosine learning rate schedule is parametrized based on the *maximum training duration* a model would see if it *survives all* SH rounds.
> $\rightarrow$ This ensures that learning curves progress identically to fully trained models, even under early pruning.
>
> Note that while the Chinchilla paper may have observed sensitivity to LR schedules, our approach preserves the intended schedule for each model individually, so the SH+surrogate selection does not alter the actual underlying learning dynamics.
>
> ---
> ---
> We hope our answers have addressed all your questions and concerns.
> If you have any further queries, please let us know and we will do our best to promptly clarify any remaining points.

---

### Author Response · Authors · 2025-12-04
**Summary of Rebuttal and Revisions**

We would like to again thank the Reviewers and AC for their constructive feedback.
This paper proposes a compute-efficient methodology for active budget allocation to fit scaling laws, and we have revised the manuscript to clarify this scope and further improve contextualization. We summarize our main responses below:

**Contextualization & Related Work (AC *AGqn*):**
Following the AC's recommendation, we explicitly positioned our work against 'observational' scaling laws (e.g., Ruan et al., Polo et al.).
$\Rightarrow$ This reinforces the distinction that our method focuses on **active budget allocation** to generate data, whereas observational methods rely on existing checkpoints or benchmark results (*extended Section 2, highlighted in blue*).

**Practicality & Implementation (*t952, qfWJ*):**
- **Negligible Overhead:** The Successive Halving (SH) + Surrogate steps introduce negligible cost compared to model training and integrate cleanly into distributed workflows without complex stop-restart overhead.
- **Strong Extrapolation:** As shown in Table 3 and Figure 5, the method successfully extrapolates to unseen larger models and budgets, preserving the true scaling relationship beyond the training data.
- **Preserved Dynamics:** Cosine schedules are parametrized to the *maximum possible duration*. This ensures that SH pruning does not alter the underlying learning curve shapes or training dynamics compared to standard runs.

**Methodological Consistency (*2jwe, qfWJ*):**
- **Modular Surrogates:** Our framework is modular, allowing practitioners to swap surrogate models (e.g., Deep Ensembles vs. GPs) based on the required fidelity or task knowledge.
- **Budget Constraints:** Reported optima represent the valid trade-offs within a *fixed, limited data budget* (10B tokens). This is internally consistent for resource-constrained research and distinct from the 'unlimited data' assumptions of the Chinchilla regime.

**Clarification of Scope (Addressing *jZTL*):**
We address a fundamental misunderstanding regarding the core goals of our work raised by Reviewer *jZTL*:
- **Goal:** The reviewer seems to evaluate our work as a *characterization* of scaling behavior (critiquing functional forms and 'strawman' baseline).
  $\Rightarrow$ Our contribution is a *compute-efficient methodology* to generate the data points required to fit *existing* laws, not the proposal of new functional forms.
- **Baselines:** Consequently, 'uniform allocation' (equally or fully training all candidates) is the principled baseline for quantifying efficiency gains, rather than a 'strawman.'
- **Proxy Task:** The proxy objective functions to rank and prune the candidate *set* to generate a *set* of curves under given resource constraints, rather than selecting a single final model.


$\Rightarrow$ We hope these clarifications better contextualize the scope of our work and reinforce our primary contribution: *a robust, budget-efficient strategy for deriving scaling laws*.

---

### Meta-Review · Area_Chair_mTDs · 2026-01-06

**Summary:**

All the reviewers acknowledged the importance of the problem being tackled by the paper, as well as the remarkable cost reduction achieved by the proposed method. The major concerns include -

1. Unclear overhead introduced by the proposed approach (Reviewer t952)
2. Potential suboptimal extrapolative prediction performance (Reviewer t952)
3. Some designs of the approach, including the surrogate model and noise model, may be imprecise and introduce errors (Reviewers 2jwe and qfWJ)
4. Only the validation loss is being modeled, but no other generalization metrics (Reviewer 2jwe)
5. Experiments were done on limited models (NanoGPT) (Reviewer 2jwe)
6. Suboptimal hyperparameters (Reviewer qjWJ)
7. Contradiction with Chinchilla's conclusion on compute-optimal frontier (Reviewer qfWJ)
8. Insufficient writing clarity; the main text is not self-sufficient (Reviewer jzTL)
9. The paper does not seem to meet the goals of common scaling laws, either extrapolative prediction or comparing two model design choices at a large scale (Reviewer jzTL)

**Reviewer Concerns:**

Most concerns are adequately addressed.

1. The authors have argued that the overhead introduced by the approach is negligible compared to the computational savings it achieves, which is a valid argument.
2. The authors pointed out that the extrapolative prediction performance is already reported in the paper.
3. Regarding the surrogate model, the authors adopt a modular approach to attempt multiple surrogate models, which alleviates the problem. Regarding the noise model, the authors argue that their intent is to choose multiple noise models that collectively represent a diverse set of noise behaviors, rather than fitting to a particular noise pattern.
4. The authors mentioned that they will pursue other generalization losses as a future direction.
5. The authors mentioned that they do not have sufficient resources to carry out larger-scale experiments.
6. The authors explained that they chose a single hyperparameter setting to keep the comparison controlled.
7. The authors clarified that the proposed method works under limited resources, which is a different setting from Chinchilla's paper.
8. The authors are open to adding some key details back to the paper.
9. The authors clarified that the goal of the paper is to derive a resource-efficient way to train the models for the estimation of the scaling laws, which is orthogonal and compatible with either of the said scaling law goals.

In short, due to the compute resource limitations, the authors are unable to extend the experiments to more models, or combine the proposed method with other scaling law designs along the orthogonal directions. While this is completely understandable, this does impact the quality and scope of the paper. Also, according to Reviewer jzTL, the paper needs a major restructuring to rearrange the details that appear in the main text, particularly with the revised position.

**Reviewer Scores:**

Reviewer t952 may increase their score to 6.

Reviewer 2jwe and qfWJ are likely to maintain their score to 6.

Reviewer jzTL may increase their score to 4, with the reservation that the writing quality is uncertain.

After the score adjustments, the paper becomes a borderline paper, and the decision on this paper is hard. Considering that this paper can be greatly improved should there be more time (for running new experiments under the limited compute budget, and for improving the writing), I would recommend reject.

---

### Decision · Program_Chairs · 2026-01-26

Reject